# Symptom burden and health-related quality of life in chronic kidney disease: A global systematic review and meta-analysis

Benjamin R. Fletcher[1], Sarah Damery[2], Olalekan Lee Aiyegbusi[1,2,3], Nicola Anderson[1,4], Melanie Calvert[1,3,5], Paul Cockwell[1,4,6], James Ferguson[1,3], Mike Horton[7], Muirne C. S. Paap[8], Chris Sidey-Gibbons[9], Anita Slade[1,3,5], Neil Turner[10], Derek Kyte[1,11]*

1 Centre for Patient Reported Outcomes Research, Institute of Applied Health Research, University of Birmingham, Birmingham, United Kingdom, 2 National Institute for Health Research Applied Research Collaboration West Midlands, Institute of Applied Health Research, University of Birmingham, Birmingham, United Kingdom, 3 National Institute for Health Research Birmingham Biomedical Research Centre, University of Birmingham, Birmingham, United Kingdom, 4 Department of Renal Medicine, Queen Elizabeth Hospital Birmingham, University Hospitals Birmingham, Birmingham, United Kingdom, 5 National Institute for Health Research Surgical Reconstruction and Microbiology Research Centre, University of Birmingham, Birmingham, United Kingdom, 6 Institute of Inflammation and Ageing, University of Birmingham, Birmingham, United Kingdom, 7 Psychometric Laboratory for Health Sciences, University of Leeds, Leeds, United Kingdom, 8 Department of Child and Family Welfare, Faculty of Behavioural and Social Sciences, University of Groningen, Groningen, The Netherlands, 9 MD Anderson Center for INSPiRED Cancer Care, University of Texas, Houston, Texas, United States of America, 10 Centre for Inflammation Research, University of Edinburgh, Edinburgh, United Kingdom, 11 School of Allied Health and Community, University of Worcester, Worcester, United Kingdom

* d.kyte@worc.ac.uk

**Data Availability Statement:** All relevant data are within the manuscript and its Supporting information files.

## Abstract

### Background

The importance of patient-reported outcome measurement in chronic kidney disease (CKD) populations has been established. However, there remains a lack of research that has synthesised data around CKD-specific symptom and health-related quality of life (HRQOL) burden globally, to inform focused measurement of the most relevant patient-important information in a way that minimises patient burden. The aim of this review was to synthesise symptom prevalence/severity and HRQOL data across the following CKD clinical groups globally: (1) stage 1–5 and not on renal replacement therapy (RRT), (2) receiving dialysis, or (3) in receipt of a kidney transplant.

### Methods and findings

MEDLINE, PsycINFO, and CINAHL were searched for English-language cross-sectional/ longitudinal studies reporting prevalence and/or severity of symptoms and/or HRQOL in CKD, published between January 2000 and September 2021, including adult patients with CKD, and measuring symptom prevalence/severity and/or HRQOL using a patient-reported outcome measure (PROM). Random effects meta-analyses were used to pool data, stratified by CKD group: not on RRT, receiving dialysis, or in receipt of a kidney transplant. Methodological quality of included studies was assessed using the Joanna Briggs Institute

**Funding:** This review was funded as part of a study funded by Kidney Research UK (Stoneygate Research Award KS_RP_013_20180914; www. kidneyresearchuk.org). The grant was awarded to DK, and funded BF full time. The funders did not play any role in study design, data collection/ analysis, decision to publish, or preparation of the manuscript.

**Competing interests:** I have read the journal's policy and the authors of this manuscript have the following competing interests: MC is Director of the Birmingham Health Partners Centre for Regulatory Science and Innovation, Director of the Centre for the Centre for Patient Reported Outcomes Research and is a National Institute for Health Research (NIHR) Senior Investigator. MC receives funding from the National Institute for Health Research (NIHR) Birmingham Biomedical Research Centre, the NIHR Surgical Reconstruction and Microbiology Research Centre and NIHR ARC West Midlands at the at the University of Birmingham and University Hospitals Birmingham NHS Foundation Trust NIHR/UKRI., Health Data Research UK, Innovate UK (part of UK Research and Innovation), Macmillan Cancer Support, UCB Pharma, Janssen, GSK and Gilead. MC has received personal fees from Astellas Aparito Ltd, CIS Oncology, Takeda, Merck, Daiichi Sankyo, Glaukos, GSK and the Patient-Centered Outcomes Research Institute (PCORI) outside the submitted work. OLA is supported by the NIHR Birmingham BRC, West Midlands, Birmingham; UKRI, Janssen, Gilead and GSK. He declares personal fees from Gilead Sciences Ltd, Merck and GSK outside the submitted work. DK reports grants from Macmillan Cancer Support, Innovate UK, the NIHR, NIHR Birmingham Biomedical Research Centre, and NIHR SRMRC at the University of Birmingham and University Hospitals Birmingham NHS Foundation Trust, and personal fees from Merck outside the submitted work.

**Abbreviations:** BDI-II, Beck Depression Inventory; CKD, chronic kidney disease; EQ-5D, EuroQoL–5 Dimension; FDR, false discovery rate; HADS, Hospital Anxiety and Depression Scale; HRQOL, health-related quality of life; KDQOL, Kidney Disease Quality of Life; KSS, Kidney Disease Quality of Life Summary Score; MCS, Mental Component Summary; PCS, Physical Component Summary; PROM, patient-reported outcome measure; RRT, renal replacement therapy; SF-12, 12-Item Short Form Health Survey; SF-36, 36-Item Short Form Health Survey.

Critical Appraisal Checklist for Studies Reporting Prevalence Data, and an exploration of publication bias performed. The search identified 1,529 studies, of which 449, with 199,147 participants from 62 countries, were included in the analysis. Studies used 67 different symptom and HRQOL outcome measures, which provided data on 68 reported symptoms. Random effects meta-analyses highlighted the considerable symptom and HRQOL burden associated with CKD, with fatigue particularly prevalent, both in patients not on RRT (14 studies, 4,139 participants: 70%, 95% CI 60%–79%) and those receiving dialysis (21 studies, 2,943 participants: 70%, 95% CI 64%–76%). A number of symptoms were significantly ($p < 0.05$ after adjustment for multiple testing) less prevalent and/or less severe within the post-transplantation population, which may suggest attribution to CKD (fatigue, depression, itching, poor mobility, poor sleep, and dry mouth). Quality of life was commonly lower in patients on dialysis (36-Item Short Form Health Survey [SF-36] Mental Component Summary [MCS] 45.7 [95% CI 45.5–45.8]; SF-36 Physical Component Summary [PCS] 35.5 [95% CI 35.3–35.6]; 91 studies, 32,105 participants for MCS and PCS) than in other CKD populations (patients not on RRT: SF-36 MCS 66.6 [95% CI 66.5–66.6], $p = 0.002$; PCS 66.3 [95% CI 66.2–66.4], $p = 0.002$; 39 studies, 24,600 participants; transplant: MCS 50.0 [95% CI 49.9–50.1], $p = 0.002$; PCS 48.0 [95% CI 47.9–48.1], $p = 0.002$; 39 studies, 9,664 participants). Limitations of the analysis are the relatively few studies contributing to symptom severity estimates and inconsistent use of PROMs (different measures and time points) across the included literature, which hindered interpretation.

## Conclusions

The main findings highlight the considerable symptom and HRQOL burden associated with CKD. The synthesis provides a detailed overview of the symptom/HRQOL profile across clinical groups, which may support healthcare professionals when discussing, measuring, and managing the potential treatment burden associated with CKD.

## Protocol registration

PROSPERO CRD42020164737.

## Author summary

### Why was this study done?

- Chronic kidney disease (CKD) is a common disease globally.
- Patients with CKD have a reduced quality of life and a greater risk of hospitalisation, heart problems, and death.
- Monitoring patient symptoms and quality of life can provide important information to help optimise CKD management.
- There is a lack of clear evidence on differences in patient quality of life between CKD groups and which symptoms are experienced most often and/or are most severe.

## What did the researchers do and find?

- We reviewed 449 studies that included 199,147 patients from 62 countries.

- Patients with CKD reported a range of common and/or severe symptoms; the exact symptom burden depended on the stage of the disease and how it was being treated. Fatigue, however, was a very common and severe symptom in all patient groups.

- Quality of life for patients with CKD was significantly lower than for individuals without the disease, and was worst in patients receiving dialysis.

- In general, patients who had received a kidney transplant experienced fewer and less severe symptoms and had an improved quality of life, but this was still worse than that of people without CKD.

## What do these findings mean?

- Symptom burden and negative impact on quality of life are considerable for people with CKD, especially for those receiving dialysis treatment.

- The findings of this study will support healthcare professionals when discussing, measuring, and managing the potential treatment burden associated with CKD.

- This global review of symptoms in patients with CKD will help in the selection of symptoms for inclusion in remote monitoring to identify patients in need of intervention.

## Introduction

Chronic kidney disease (CKD) has an estimated global prevalence of 9.1% (700 million people) and is associated with a major increased risk of early death for those affected, with 4.6% of deaths annually attributable to impaired kidney function [1]. In addition, CKD represents a substantial health economic burden, with advancement from stage 3 to 4/5 associated with a 1.3-to 4.2-fold increase in costs, and progression to end-stage renal disease (ESRD) estimated to cost $20,000–$100,000 per patient per year [2].

Patients with CKD experience an increased risk of hospitalisation and mortality [3], and reduced health-related quality of life (HRQOL), which is independently associated with cardiovascular disease events and death [4–6]. Patient-reported outcomes, including HRQOL and symptoms, are often identified by patients with CKD as more important to them than clinical outcomes such as survival [7].

There has been an increasing move towards models of remote and virtual care for patients with CKD [8], accelerated by the emergence of COVID-19, within which, capture of symptom and HRQOL data are seen as key adjuncts to support optimal care [9–11]. However, there remains a lack of research that has synthesised global data on CKD-specific symptom and HRQOL burden to inform collection of the most relevant patient-important data in a way that minimises patient burden.

The aim of this study was to (1) produce a comprehensive and consolidated global synthesis of symptom prevalence/severity and HRQOL across CKD treatment groups, (2) explore which

symptom/HRQOL domains are modified by CKD and may be attributable to the disease, and (3) determine which patient-reported outcome measures (PROMs) are currently available to capture symptom prevalence/severity and HRQOL in CKD.

## Methods

The Preferred Reporting Items for Systematic Reviews and Meta-Analyses (PRISMA) guidelines were followed (S1 Appendix), and the study protocol was registered with PROSPERO (CRD42020164737) (S9 Appendix). The protocol and analysis methods were developed prospectively.

### Search strategy and data sources

The following databases were searched from 1 January 2000 until 6 February 2020 (searches updated on 1 September 2021): Ovid MEDLINE, Ovid PsycINFO, and EBSCO CINAHL (for the full search strategy, see S2 Appendix). Two authors (2 of BRF, SD, and DK) independently assessed selected articles for eligibility at the title/abstract and full-text stages, with disagreements resolved through discussion.

### Inclusion and exclusion criteria

Studies were included if they met the following criteria: (1) published in or after January 2000; (2) written in English; (3) included adult (≥18 years) patients with CKD at stage 1–5 not receiving renal replacement therapy (RRT), those receiving dialysis, or those in receipt of a kidney transplant; (4) reported prevalence and/or severity of symptoms and/or HRQOL measured using a PROM; and (5) used a cross-sectional or longitudinal study design. Studies were excluded if they (1) were editorials, conference abstracts, reports of qualitative findings, or systematic reviews; (2) solely reported symptoms or HRQOL that were not self-reported by patients (e.g., clinician-reported); or (3) were not reported in English.

### Data extraction

The following data were independently extracted into a pre-piloted spreadsheet by 2 authors (2 of BRF, SD, and DK) and disagreements resolved through discussion: study information (year conducted, country of origin, single/multi-centre, cross-sectional/longitudinal design), study population (inclusion/exclusion criteria, CKD stage, estimated glomerular filtration rate [eGFR], co-morbidity indices, demographics), and study outcomes (measures used, symptom prevalence and severity, HRQOL PROM scores). Where possible, we also attempted to extract data from the included studies that were collected from contemporaneous (ideally matched) non-CKD control populations.

### Quality appraisal

Methodological quality of included studies was assessed using the Joanna Briggs Institute Critical Appraisal Checklist for Studies Reporting Prevalence Data [12]. Studies were assessed for adequacy of sampling (frame and method), sample size and description, data analysis, and comparability of outcomes across studies. Two authors independently conducted the appraisal (BRF and SD), and disagreements were resolved through discussion, or by a third reviewer (DK) where required.

## Data analysis

We followed established guidelines for systematic reviews of observational epidemiological studies reporting prevalence with a focus on estimating the global burden of disease [12]. As outlined in the protocol, prevalence/severity data were pooled using either a random or fixed effects model depending on the heterogeneity of the included studies. Heterogeneity was determined using Cochran's $Q$ test at a significance level of 0.10. Heterogeneity was quantified using the $I^2$ statistic (acceptable heterogeneity defined as $I^2 < 70\%$) [13]. All analyses had high heterogeneity; therefore, a random effects model was used. Subgroup analysis was performed based on the stage of CKD (categorised as not on RRT, on dialysis, or in receipt of a kidney transplant), if there were 3 or more studies within a subgroup. Publication bias was assessed where meta-analyses included 10 or more studies using Egger's test for funnel plot asymmetry, with a significance level of $p < 0.05$ [14]. Using Stata, the metaprop (meta-analysis of binomial data) command was used to summarise prevalence data, and the metan command was used to summarise severity and HRQOL scores [15,16].

To aid comparison of symptom severity data provided across different outcome measures, all mean severity scores were converted to a 0–100 scale, where a higher score indicates greater severity. For HRQOL scores, 100 represents the best possible quality of life. For example, the Beck Depression Inventory (BDI-II) scale results in a severity score of 0–63; therefore, a score of 43 would convert to 68.3 on a 0–100 scale: $43/63 \times 100$. Symptom severity scores were also combined using random effects meta-analysis.

A weighted composite summary score for the Kidney Disease Quality of Life (KDQOL) instrument—KDQOL Summary Score (KSS)—was calculated by combining the 'symptoms and problems' (12 items), 'effects of kidney disease' (8 items), and 'burden of kidney disease' (4 items) domains. This summary score was calculated using the recommended method reported by Peipert et al. [17], in which mean scores and standard deviations for each of the 3 domains were combined, weighted by the number of items per domain.

Presentation of symptom prevalence and severity focused on 2 areas: those symptoms that were most prevalent/severe across populations and those symptoms that were significantly different between populations not receiving RRT and those receiving dialysis or transplantation. The latter area is important, as it could provide insight into those symptoms that may be attributable to changes in renal function and may provide potential targets for symptom tracking in CKD.

Exploratory subgroup analysis was used to compare prevalence and score (severity and HRQOL) estimates between groups in meta-analyses (not on RRT versus dialysis, not on RRT versus transplant, and dialysis versus transplant). To account for multiple testing, sharpened false discovery rate (FDR) $q$-values were computed [18], and adjusted $p$-values are reported, with a significance level of $p < 0.05$. All analyses were conducted using Stata (version 15.0).

## Role of the funder

The funder of the study had no role in study design, data collection, data analysis, data interpretation, or writing of the report. The corresponding author had full access to all the data in the report and had final responsibility for the decision to submit for publication.

## Results

### Included studies

Searches identified 1,521 records after deduplication, and an additional 8 were identified through citation searches of included studies. Following title/abstract screening, 631 full-text articles were obtained, with 182 excluded at this stage, leaving 449 studies for inclusion in the final syntheses (Table 1). Information on individual studies included in this review (outcomes

**Table 1. Characteristics of the included 449 studies.**

| Characteristic | Number of studies (%) |
|---|---|
| **Publication year** | |
| 2000–2005 | 44 (10%) |
| 2006–2010 | 88 (19%) |
| 2011–2015 | 126 (28%) |
| 2016–2021 | 191 (43%) |
| **Population (stage of CKD)** | |
| Not on RRT | 126 (28%) |
| Stage 1 | 29 (6%) |
| Stage 2 | 44 (10%) |
| Stage 3 | 80 (18%) |
| Stage 4 | 92 (20%) |
| Stage 5 | 98 (22%) |
| Dialysis | 274 (61%) |
| Haemodialysis | 228 (51%) |
| Peritoneal dialysis | 118 (26%) |
| Kidney transplant | 139 (31%) |
| **Country** | |
| Total* | 62 |
| US | 43 (10%) |
| Brazil | 43 (10%) |
| UK | 36 (8%) |
| Turkey | 30 (7%) |
| China | 29 (6%) |
| South Korea | 24 (5%) |
| Australia | 18 (4%) |
| Netherlands | 17 (4%) |
| Spain | 16 (4%) |
| Italy | 15 (3%) |
| Taiwan | 15 (3%) |
| Japan | 13 (3%) |
| Canada | 12 (3%) |
| Iran | 11 (2%) |
| Germany | 10 (2%) |
| **Recruitment** | |
| Single centre | 251 (56%) |
| More than 1 study centre | 181 (40%) |
| Not reported/unclear | 17 (4%) |
| **Outcomes** | |
| Symptoms | 181 (40%) |
| Health-related quality of life | 361 (80%) |
| **Study design** | |
| Cross-sectional | 385 (86%) |
| Longitudinal | 64 (14%) |

CKD, chronic kidney disease; RRT, renal replacement therapy.

*Only countries with 10 studies or more listed; full list available in S3 Appendix.

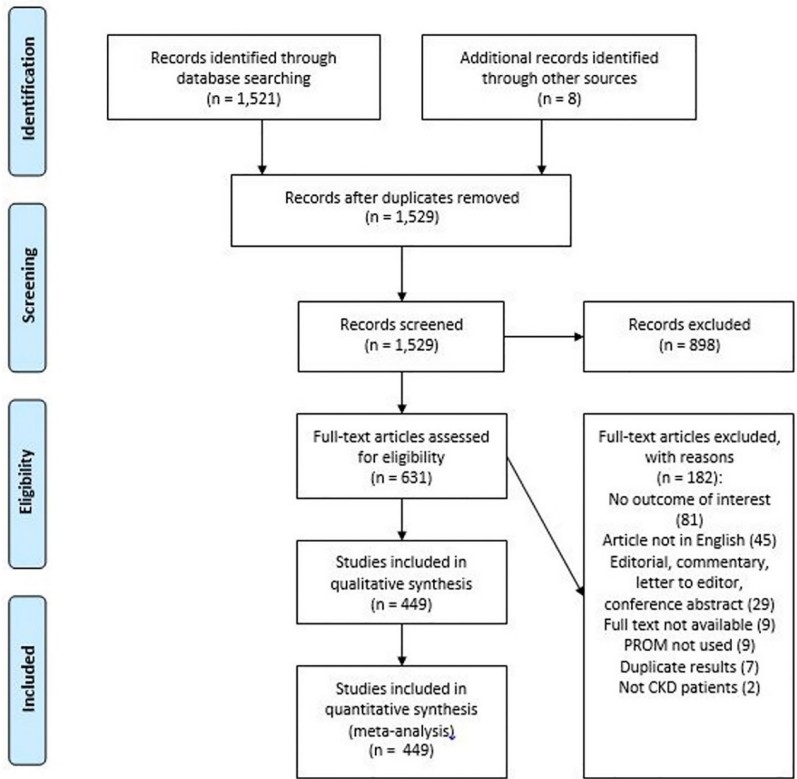

**Fig 1. PRISMA diagram.** CKD, chronic kidney disease; PROM, patient-reported outcome measure.

used, study design, country of origin, population, and risk of bias) is included in S3 Appendix. The full lists of included and excluded studies are provided in S4 and S5 Appendices, respectively. The PRISMA diagram is shown in Fig 1.

There was a total of 199,147 participants involved in the included studies (median 146, IQR 85 to 267, range 9 to 18,015). Studies were conducted in 62 countries, with the most studies in the following countries: US, 43 (10%); Brazil, 43 (10%); UK, 36 (8%); Turkey, 30 (7%); and China, 29 (6%); the majority of studies were conducted at a single centre (251, 56%). Most studies were cross-sectional in design (385, 86%). Patients with CKD stage 1–5 who were not on RRT were included in 126 (28%) studies. The staging of patients not receiving RRT was as follows: 29 studies, stage 1; 44 studies, stage 2; 80 studies, stage 3; 92 studies, stage 4; and 98 studies, stage 5. Patients receiving dialysis were included in 274 (61%) studies (explicitly stated as haemodialysis in 228, and peritoneal dialysis in 118). Patients in receipt of a kidney transplant were included in 139 (31%) studies.

## Outcome measures

The included studies utilised 67 different PROMs to collect information on symptoms and HRQOL. Eleven measures were reported in 10 or more individual studies: the 36-Item Short Form Health Survey (SF-36) in 227 studies, KDQOL in 100 studies, the 12-Item Short Form Health Survey (SF-12) in 52 studies, the BDI-II in 51 studies, the World Health Organization Quality of Life (WHOQOL-BREF) instrument in 33 studies, EuroQoL–5 Dimension (EQ-5D) in 28 studies, the Hospital Anxiety and Depression Scale (HADS) in 22 studies, the Centre for Epidemiologic Studies Depression Scale (CES-D) in 11 studies, the Patient Health

Questionnaire–9 (PHQ-9) in 10 studies, the Integrated Palliative care Outcome Scale–Renal (IPOS-Renal) in 10 studies, and the Pittsburgh Sleep Quality Index (PSQI) in 10 studies.

A total of 68 different symptoms were measured across 54 PROMs (mean number of items per PROM = 22, range 1–90). No single PROM measured the majority of reported symptoms across the CKD population. The PROMs with the most comprehensive symptom coverage included the CKD Symptom Burden Index (44% of symptoms), the Dialysis Symptom Index (41%), the Memorial Symptom Assessment Scale–Short Form (33%), the Modified Transplant Symptom Occurrence and Symptom Distress Scale (33%), and the Chronic Kidney Disease Symptom Index (32%). There was little consistency across measures; some focused on a single symptom (e.g., BDI-II: depression), others included a number of symptom subdomains (e.g., HADS: anxiety and depression), and some included multiple questions, each tackling a different symptom (e.g., Disease Symptom Index: 30 individual symptom questions).

## Symptom prevalence and severity

Data on symptom prevalence and severity were extracted from 181 studies. Pooled summary data are available in Table 2. Symptom prevalence data were available for 45 symptoms in patients not on RRT, for 42 symptoms in patients receiving dialysis, and for 27 symptoms in the transplant population. Symptom severity data were available for 18 symptoms in patients not on RRT, for 33 symptoms in patients receiving dialysis, and for 22 symptoms in transplant patients. Data for symptom prevalence and severity are shown in Figs 2–7.

Patients not on RRT and those receiving dialysis shared a similar profile of the most prevalent symptoms. For example, symptoms with a reported prevalence of >50% in both populations included fatigue and poor mobility, and symptoms with a reported prevalence of >45% in both populations included bone/joint pain, general pain, poor sleep, sexual dysfunction, heartburn, muscle cramps, itching, and dry skin. Fewer data were available for transplant patients; however, indigestion, abdominal pain, constipation, muscle weakness, and muscle cramps were most prevalent, being present in >50% of patients.

For patients not on RRT, the most severe symptoms included sexual dysfunction, anxiety, itching, and depression. Pain was the most severe symptom in dialysis, followed by fatigue, dry skin, and bone/joint pain. For transplant recipients, the most severe symptoms were change in appearance, blurred vision, and excessive appetite.

## Trends in prevalence/severity across clinical groups and attribution

Within the included studies, data from contemporaneous non-CKD control populations were limited, and available for only 17 symptoms for prevalence and 4 for severity.

Prevalence was higher in CKD patients compared to healthy controls for 14 of 17 symptoms (bone/joint pain, fatigue, trouble with memory, muscle cramps, itching, restless legs, muscle weakness, constipation, shortness of breath, anxiety, depression, decreased appetite, diarrhoea, and abdominal pain), and lower than controls for 1 symptom (stress).

Fatigue was the most prevalent symptom in patients not on RRT and in those on dialysis. Fatigue was also the second most severe symptom in dialysis patients (adjusted severity score 51.5, 95% CI 29.1–33.8). Fig 8 displays the full results of the meta-analysis of fatigue prevalence across CKD clinical groups including controls. Fatigue prevalence in controls was 34% (95% CI 0%–70%). In comparison, fatigue prevalence was significantly higher (FDR-sharpened $q$-value 0.021) in patients with CKD not on RRT (70%, 95% CI 60%–79%) and in dialysis patients (70%, 95% CI 64%–76%; FDR-sharpened $q$-value 0.021); fatigue prevalence was significantly lower in transplant patients (48%, 95% CI 32%–63%; FDR-sharpened $q$-value 0.005) than in patients on RRT or dialysis, although notably not as low as in controls. A number of

**Table 2. Prevalence and severity of symptoms across CKD populations.**

| Symptom | Measures | Not on RRT | | | | Dialysis | | | | Transplant | | | |
|---|---|---|---|---|---|---|---|---|---|---|---|---|---|
| | | Prevalence, percent (95% CI) | Number of studies (participants) | Severity* (95% CI) | Number of studies (participants) | Prevalence, percent (95% CI) | Number of studies (participants) | Severity* (95% CI) | Number of studies (participants) | Prevalence, percent (95% CI) | Number of studies (participants) | Severity* (95% CI) | Number of studies (participants) |
| Fatigue | 9, 10, 11, 14, 15, 17, 22, 25, 27 | 70 (60–79) | 14 (4,139) | 22.8 (18.7–26.8) | 1 (143) | 70 (64–76) | 21 (2,943) | 51.1 (47.2–65.3) | 8 (1,181) | 48 (32–63) | 8 (1,938) | 56.1 (47.2–56) | 8 (1,021) |
| Poor mobility | 27 | 56 (34–77) | 4 (303) | 19 (14.5–23.5) | 1 (143) | 55 (49–61) | 2 (240) | | | 13 (7–20) | 1 (110) | | |
| Bone or joint pain | 10, 11, 14, 23 | 55 (40–71) | 8 (3,993) | | | 49 (36–62) | 12 (1,659) | 37.9 (28.6–47.1) | 3 (447) | 55 (49–62) | 1 (230) | | |
| Drowsiness | 10, 11, 25, 39 | 53 (40–67) | 7 (448) | 22.5 (18.4–26.6) | 1 (143) | 34 (20–48) | 6 (1,066) | | | 47 (42–52) | 2 (362) | 11.7 (11.2–12.2) | 2 (162) |
| Pain | 15, 24, 27, 34 | 53 (46–60) | 7 (503) | 22.5 (18–27) | 1 (143) | 48 (27–68) | 5 (1,054) | 55.9 (14.7–97.1) | 2 (660) | 44 (39–49) | 2 (362) | 46 (4.9–87.2) | 2 (339) |
| Poor sleep | 3, 10, 11, 12, 14, 23, 25, 27, 28 | 49 (41–58) | 15 (4,444) | 23.8 (18–29.5) | 1 (143) | 57 (52–62) | 17 (2,575) | 35 (28.9–40.9) | 6 (1,917) | 31 (14–48) | 6 (1,529) | 30 (22.3–37.6) | 3 (404) |
| Sexual dysfunction | 2, 10, 11, 14, 16, 21, 23, 25, 42 | 48 (35–62) | 11 (4,080) | 56.4 (32–80.8) | 3 (74) | 49 (40–58) | 16 (1,671) | 36.8 (26.7–46.8) | 5 (529) | 38 (27–49) | 4 (402) | 60.9 (46.7–75.1) | 1 (20) |
| Itching | 1, 10, 11, 14, 25, 35, 38 | 46 (38–55) | 15 (4,208) | 25 (8.2–41.8) | 1 (87) | 51 (41–60) | 19 (3,571) | 35.7 (29.2–42.3) | 9 (1,863) | 30 (21–38) | 3 (473) | | |
| Heartburn | 11 | 46 (43–48) | 2 (1,161) | | | 66 (49–80) | 1 (38) | | | 50 (43–62) | 1 (230) | | |
| Muscle cramps | 10, 11, 14 | 46 (30–62) | 7 (3,710) | | | 53 (43–62) | 12 (1,569) | 25.1 (21.6–28.6) | 2 (330) | | | | |
| Dry skin | 10, 11, 14 | 45 (33–57) | 6 (3,263) | | | 57 (51–64) | 9 (884) | 42.1 (15.3–68.8) | 2 (330) | | | | |
| Swelling in legs | 10, 11, 14, 25 | 45 (36–53) | 8 (3,340) | | | 39 (29–50) | 9 (811) | | | | | | |
| Worrying | 10, 14 | 44 (37–50) | 4 (2,102) | | | 44 (31–57) | 8 (846) | 32.3 (28.8–35.8) | 2 (330) | 53 (46–60) | 1 (230) | | |
| Muscle weakness | 23 | 43 (20–66) | 3 (1,166) | | | 68 (64–71) | 2 (658) | | | | | | |
| Decreased appetite | 10, 11, 14, 15, 23, 24, 25, 27 | 42 (32–52) | 16 (8,408) | 19.8 (14.8–24.7) | 1 (143) | 40 (27–54) | 16 (2,576) | 24.6 (1.6–47.7) | 2 (821) | 28 (24–32) | 3 (592) | 7 (4–10) | 1 (134) |
| Shortness of breath | 10, 11, 14, 15, 23, 25, 27 | 42 (34–51) | 15 (4,809) | 15 (11.3–18.7) | 1 (143) | 38 (28–48) | 16 (2,593) | 27.7 (20.3–35.2) | 3 (938) | 40 (36–44) | 3 (592) | 9 (5.6–12.4) | 1 (134) |
| Dry mouth | 10, 14, 25 | 41 (33–49) | 12 (2,657) | 12.5 (8.8–16.2) | 1 (143) | 43 (35–52) | 11 (1,203) | 33.6 (16–51.3) | 3 (447) | 41 (35–48) | 1 (110) | | |
| Sleep apnoea | 6 | 40 (23–59) | 1 (30) | | | 34 (19–53) | 1 (32) | | | | | | |
| Feeling irritable | 10, 11, 14 | 39 (17–62) | 6 (3,263) | | | 46 (27–66) | 8 (784) | 27.5 (23.9–31.1) | 1 (230) | | | | |
| Trouble with memory | 11 | 38 (15–61) | 3 (1,691) | | | 51 (30–71) | 3 (696) | | | 43 (37–50) | 1 (230) | | |
| Muscle soreness | 10, 14 | 36 (28–44) | 4 (2,102) | | | 34 (30–39) | 7 (746) | 25 (21.8–28.2) | 1 (230) | | | | |
| Constipation | 10, 14, 19, 25, 27, 30 | 35 (30–41) | 11 (2,230) | 22.5 (17.6–27.4) | 1 (143) | 32 (25–38) | 13 (1,418) | 15.4 (1.4–29.3) | 3 (1,418) | 56 (55–58) | 2 (4,342) | 11.7 (11.2–12.2) | 1 (4,232) |
| Feeling sad | 10, 11, 14 | 34 (14–53) | 6 (3,263) | | | 40 (29–52) | 8 (784) | 25 (21.8–28.2) | 1 (230) | | | | |
| Numbness in hands/feet | 10, 11, 14, 25 | 34 (29–38) | 8 (3,340) | | | 38 (28–47) | 10 (1,001) | 36.1 (26.6–45.5) | 3 (447) | | | | |
| Dizziness | 10, 14, 25 | 33 (27–38) | 6 (2,179) | | | 36 (26–46) | 8 (846) | 22.4 (9.4–35.3) | 1 (230) | | | | |
| Changes in skin | 25, 27 | 31 (17–45) | 8 (1,541) | 10 (6.3–13.7) | 1 (143) | 45 (32–58) | 3 (278) | | | 32 (23–41) | 1 (110) | | |
| Headache | 2, 10, 14 | 31 (18–44) | 3 (2,002) | | | 30 (24–37) | 6 (659) | 27.1 (8.3–45.9) | 2 (347) | 32 (23–41) | 1 (110) | 23.3 (17.5–29.2) | 1 (100) |
| Feeling nervous | 10, 14 | 31 (25–37) | 4 (2,102) | | | 35 (18–53) | 7 (746) | 37.5 (33.6–41.4) | 1 (230) | | | | |

(Continued)

**Table 2.** (Continued)

| Symptom | Measures | Not on RRT | | | | Dialysis | | | | Transplant | | | |
|---|---|---|---|---|---|---|---|---|---|---|---|---|---|
| | | Prevalence, percent (95% CI) | Number of studies (participants) | Severity* (95% CI) | Number of studies (participants) | Prevalence, percent (95% CI) | Number of studies (participants) | Severity* (95% CI) | Number of studies (participants) | Prevalence, percent (95% CI) | Number of studies (participants) | Severity* (95% CI) | Number of studies (participants) |
| Anxiety | 4, 10, 14, 15, 20, 22, 36, 40 | 30 (23–37) | 10 (2,361) | 25.2 (14.7–35.8) | 4 (352) | 31 (24–38) | 24 (3,656) | 31.3 (26.2–36.1) | 19 (3,350) | 20 (5–34) | 5 (865) | 29.9 (24.1–35.7) | 12 (1,022) |
| Difficulty concentrating | 10, 11, 14, 23, 25, 42 | 30 (18–43) | 10 (3,976) | | | 38 (23–53) | 8 (784) | 22.5 (18.9–26.1) | 1 (230) | | | | |
| Cough | 10, 14, 25 | 29 (24–34) | 6 (2,179) | | | 31 (23–38) | 7 (746) | 20 (16.8–23.2) | 1 (230) | | | | |
| Weight loss | 11, 27 | 29 (7–50) | 3 (1,306) | | | 61 (43–76) | 1 (38) | | | | | | |
| Restless legs | 10, 11, 14, 23, 27 | 27 (21–33) | 13 (4,732) | 15 (11.3–18.7) | 1 (143) | 33 (22–43) | 13 (1,796) | 27.7 (12.7–42.8) | 2 (347) | 29 (24–34) | 2 (340) | | |
| Nausea | 11, 14, 15, 24, 27, 33 | 26 (22–30) | 12 (6,907) | 8 (5.1–10.9) | 1 (143) | 28 (20–37) | 14 (1,932) | 24.7 (12.3–37.1) | 3 (938) | 20 (16–25) | 2 (362) | 9.9 (3.7–16.1) | 2 (234) |
| Depression | 5, 7, 8, 13, 18, 20, 26, 31, 32, 37 | 26 (21–32) | 28 (14,501) | 24.6 (18.9–30.4) | 17 (10,716) | 40 (32–47) | 65 (10,253) | 29.3 (26.0–32.5) | 66 (10,210) | 24 (19–29) | 16 (4,965) | 22.2 (14.8–29.6) | 21 (3,644) |
| Bad taste in mouth | 24 | 26 (25–27) | 1 (3,599) | | | | | | | | | | |
| Feeling bloated | 2, 25 | 24 (14–33) | 2 (77) | | | | | | | 31 (22–40) | 1 (110) | | |
| Diarrhoea | 10, 11, 14, 25, 27, 30 | 17 (11–23) | 13 (3,391) | 7.5 (4.6–10.4) | 1 (143) | 20 (15–24) | 12 (1,162) | 6.5 (1–12) | 3 (524) | 24 (16–33) | 1 (110) | 17.8 (8–27.5) | 2 (4,332) |
| Sweats | 25 | 16 (8–24) | 2 (77) | | | | | | | | | | |
| Hiccups | 11 | 14 (12–16) | 2 (1,161) | | | 34 (20–51) | 1 (38) | | | | | | |
| Chest pain | 10, 14 | 14 (12–15) | 2 (1,566) | | | 14 (8–19) | 7 (776) | 25.9 (4.6–47.1) | 2 (447) | | | | |
| Vomiting | 10, 11, 14, 24, 25, 27, 33 | 14 (12–16) | 15 (7,417) | 5 (2.5–7.5) | 1 (143) | 14 (12–17) | 11 (1,141) | 27.5 (8–46.9) | 2 (447) | 11 (6–18) | 1 (110) | 13.3 (8.8–17.9) | 1 (100) |
| Change in appearance | 22, 25 | 11 (4–17) | 2 (77) | | | | | | | 11 (8–16) | 1 (274) | 79.1 (69.5–88.7) | 5 (646) |
| Stress | 29 | 11 (8–16) | 1 (236) | | | 24 (18–31) | 1 (183) | | | | | 35.7 (29.9–41.5) | 1 (100) |
| Hair loss/ growth | 25 | 5 (0–10) | 2 (77) | | | | | | | | | | |
| Indigestion | 19 | | | | | 30 (24–37) | 1 (182) | 4 (2.4–5.6) | 2 (294) | 83 (82–84) | 1 (4,232) | 20 (19.5–20.5) | 1 (4,232) |
| Abdominal pain | 19, 30 | | | | | 44 (12–75) | 4 (442) | 2.5 (0.9–4.2) | 2 (294) | 69 (68–70) | 1 (4,232) | 15 (14.5–15.5) | 1 (4,232) |
| Reflux | 19 | | | | | | | 8.3 (1.7–14.8) | 2 (294) | | | 11.7 (11.1–12.2) | 1 (4,232) |
| Blurred vision | 12 | | | | | | | | | | | 63.3 (56.1–70.5) | 1 (100) |
| Excessive appetite | 12 | | | | | | | | | | | 63.3 (56.1–70.5) | 1 (100) |
| 'Moon face' | 12 | | | | | | | | | | | 63.3 (54.8–71.8) | 1 (100) |

RRT, renal replacement therapy. Measures: (1) 5-D Pruritus, (2) Afshar 2012, (3) AIS, (4) BAI, (5) BDI-II, (6) BQ, (7) BSI, (8) CES-D, (9) CIS, (10) CKD-SBI, (11) CKD-SBI, (12) CSE, (13) DASS, (14) DSI, (15) ESAS, (16) FSFI, (17) FSS, (18) GDS 15/30, (19) GSRS, (20) HADS, (21) IIEF, (22) KTQ, (23) LUSS, (24) MDRD, (25) MSAS-SF, (26) PHQ-9, (27) IPOS-Renal, (28) PSS-4, (30) Rome II, (31) SDS, (32) TDQ, (33) TQ, (34) VAS Pain, (35) VAS Pruritus, (36) Zung SAS, (37) Zung SDS, (38) 4-Itch, (39) ESS, (40) STAI, (42) KSQ. Full names of outcome measures available in S10 Appendix.

*Severity (adjusted score), 0–100, with 100 = worst possible.

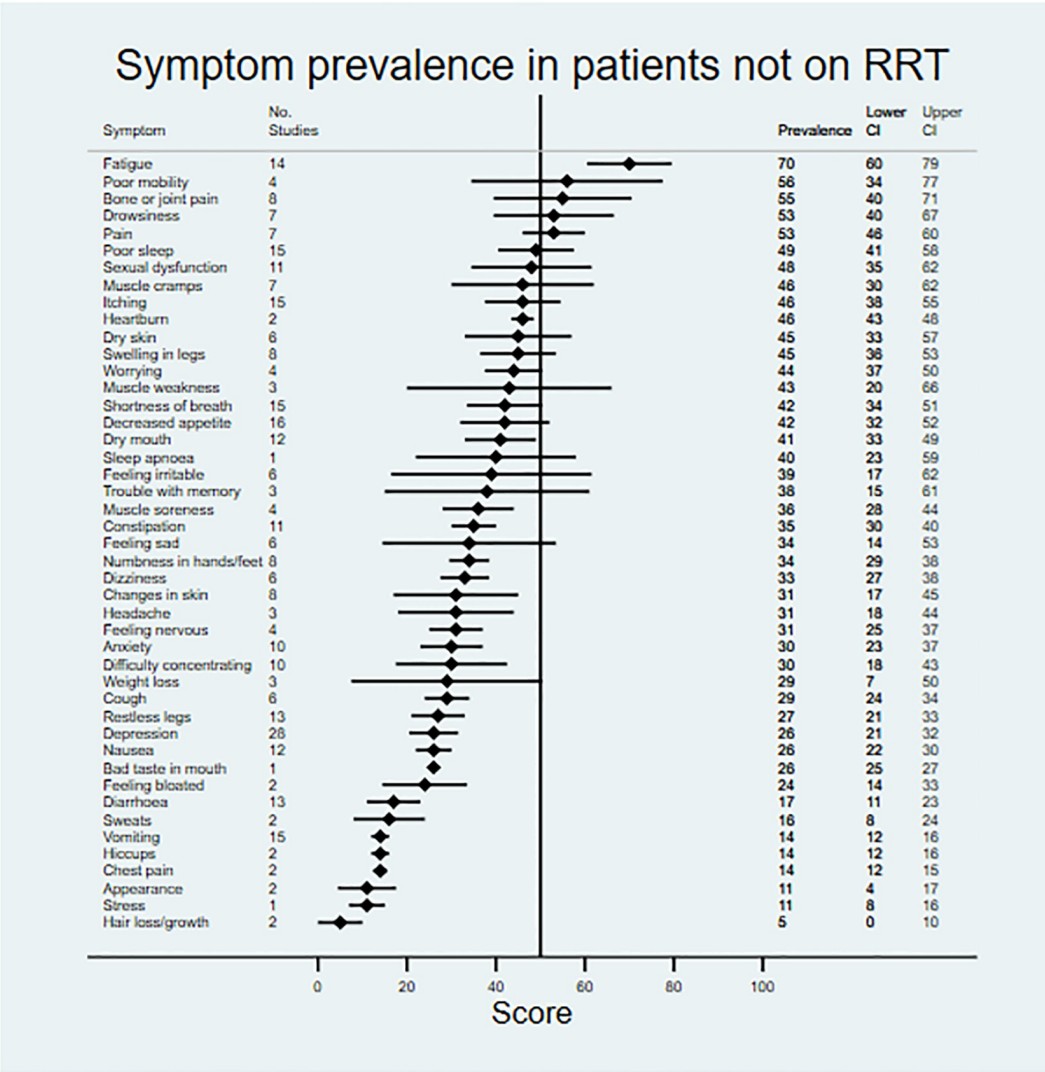

**Fig 2. Pooled prevalence of patient-reported symptoms across included studies for patients not on renal replacement therapy (RRT).**

other symptoms followed this prevalence pattern across clinical groups. All symptom prevalence and severity data are available in S6 and S7 Appendices, and all pairwise comparisons between groups including FDR-sharpened $q$-values are available in S8 Appendix. Fig 9 includes the point estimates for symptom prevalence reported across the 3 study populations (with control data where available).

Patients not on RRT and those receiving dialysis had similar profiles of prevalence across most symptoms. However, exploratory subgroup analysis (including correction for multiple testing) highlighted 2 symptoms that were significantly more prevalent in the dialysis population than in patients not on RTT: depression and stress (FDR-sharpened $q$-value 0.021 in both cases). Seven further symptoms showed tendencies towards greater prevalence (>10% difference) in the dialysis population but did not reach adjusted statistical significance: weight loss, muscle weakness, hiccups, heartburn, changes in skin, trouble with memory, and dry skin.

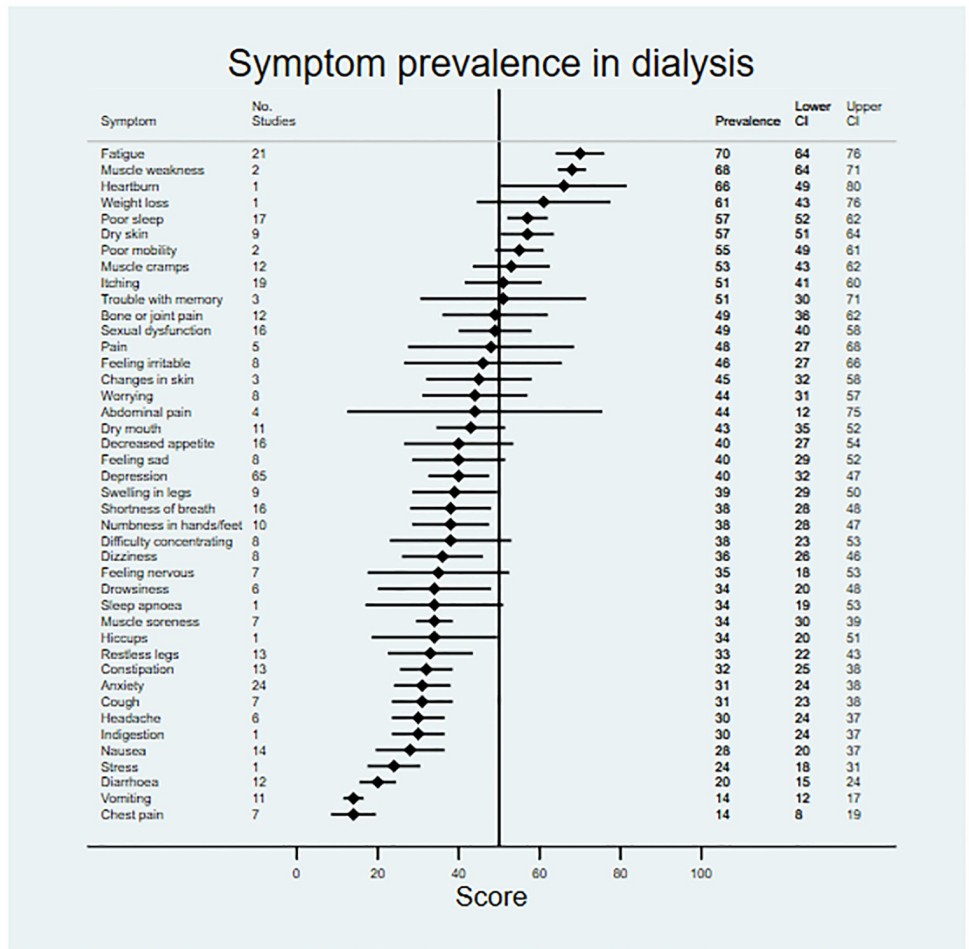

**Fig 3. Pooled prevalence of patient-reported symptoms across included studies for patients on dialysis.**

Drowsiness demonstrated a tendency towards lower prevalence in the dialysis population (>10% difference), but again the difference did not reach significance.

When compared to patients not on RRT and dialysis patients, the following symptoms were significantly less prevalent in patients who had received a kidney transplant: muscle weakness, fatigue, poor sleep, itching, decreased appetite, depression, dry mouth, and poor mobility (FDR-sharpened $q$-values 0.005–0.037). Overall, compared to the kidney transplant population, symptom prevalence was higher in patients not on RRT and patients on dialysis for 31 of 50 comparisons. However, there were 2 symptoms that reversed this pattern, constipation and indigestion, which were both significantly more prevalent in the transplant population (FDR-sharpened $q$-values 0.005–0.013).

## HRQOL

Data on HRQOL were extracted from 361 articles. The Medical Outcomes Study SF-12 and SF-36 were reported in 52 and 227 studies, respectively, KDQOL in 100, the World Health Organization Quality of Life (WHOQOL-BREF) instrument in 33, and EQ-5D in 28.

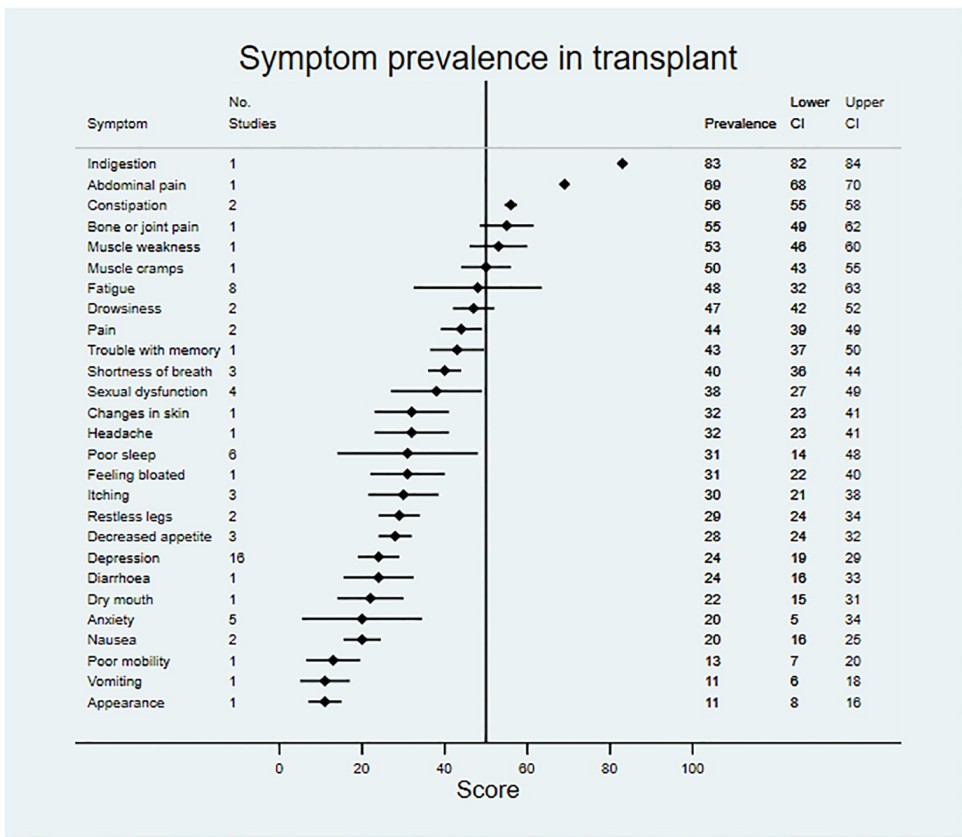

**Fig 4. Pooled prevalence of patient-reported symptoms across included studies for patients in receipt of a kidney transplant.**

Pooled scores are shown for SF-12/SF-36, KDQOL, and EQ-5D in Table 3 and Fig 10. For all scores, a higher number represents better quality of life (0–100 scale for SF-12/SF-36 and KDQOL, and possible range of −0.224 to 1 for EQ-5D).

Within the generic measures of HRQOL, SF-12/SF-36 and EQ-5D, where data were available, scores were highest in controls (EQ-5D index 0.95, 95% CI 0.95–0.95) and lowest in patients on dialysis (EQ-5D index 0.78, 95% CI 0.77–0.79; SF-36 Mental Component Summary [MCS] 45.7, Physical Component Summary [PCS] 35.5; SF-12 MCS 45.4, PCS 35.2). HRQOL scores were higher in patients receiving a kidney transplant (EQ-5D index 0.84, 95% CI 0.82–0.86; SF-36 MCS 50.0, PCS 48.0; SF-12 MCS 48.2, PCS 44.8), and higher still for patients not on RRT (EQ-5D index 0.88, 95% CI 0.88–0.88; SF-36 MCS 66.6, PCS 66.3; SF-12 MCS 49.8, PCS 47.5).

For the disease-specific KDQOL measure, the KSS was 73.0 in patients not on RRT, 64.6 in patients receiving dialysis, and highest in transplant patients (84.0). This pattern was similar in the KDQOL 'effects of kidney disease' (not on RRT, 71.7; dialysis, 63.2; transplant, 87.5) and 'burden of kidney disease' subscales (not on RRT, 50.6; dialysis, 41.7; transplant, 72.0). The burden of kidney disease subscale includes items related to how much kidney disease interferes with daily life, or makes the respondent feel like a burden. The effects of kidney disease subscale includes items exploring respondents' perceived dependency on clinicians, stress/worries, and bother associated with treatment/dietary restrictions. In the 'symptoms and problems' subscale, for patients not on RRT and those in receipt of a transplant, scores were

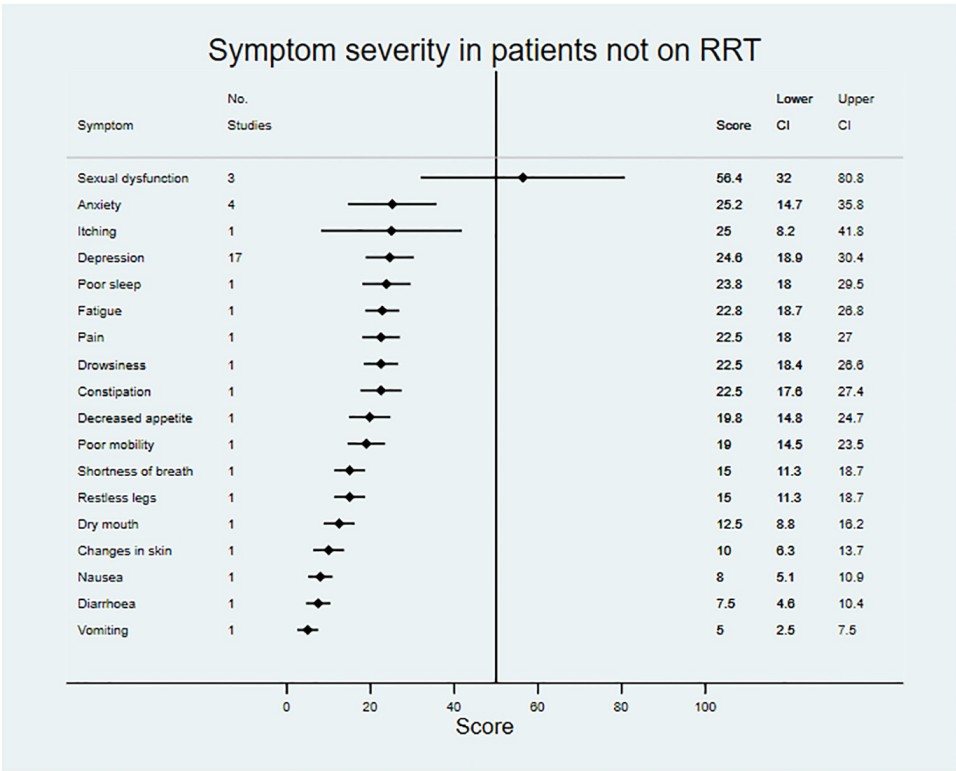

**Fig 5. Pooled symptom severity across included studies for patients not on renal replacement therapy.** Scores represent mean severity scores converted to a 0–100 scale, where a higher score indicates greater severity; vertical line at 50 for reference. RRT, renal replacement therapy.

homologous (85.9 and 86.1, respectively), whilst pooled scores for the dialysis population were lower (73.6). The symptoms and problems subscale items measured how bothered respondents were by certain symptoms (e.g., sore muscles, chest pain, cramps, itchy/dry skin, and fatigue) or problems associated with dialysis access.

All exploratory subgroup analyses comparing HRQOL scores between populations showed statistically significant differences, with the exception of the KDQOL symptoms and problems subscale comparison between patients not on RRT and in receipt of a transplant (FDR-sharpened $q$-value 0.107) (HRQOL subgroup analyses available in S8 Appendix).

## Quality appraisal of included studies

Results of the quality appraisal are shown in Fig 11 and for individual studies in S3 Appendix.

Whilst the majority of studies used random sampling or approached all patients on a clinic/registry list (59.5%), some used convenience or consecutive sampling (28.3%), or methods were unclear (12.2%). Sample size was deemed adequate in 61.9% of studies, and response rate in 57.7%. PROMs for symptoms or HRQOL that allowed comparison with other studies in CKD were used in 93.3% of studies (i.e., the measure had been used in CKD before, either as identified in this review or in the author description of previous use). Statistical analysis was reported in sufficient detail in 92.2% of studies.

There was evidence of publication bias (Egger's test for funnel plot asymmetry) for 3 of 32 symptom prevalence meta-analyses including ≥10 studies; the 3 symptoms were poor sleep,

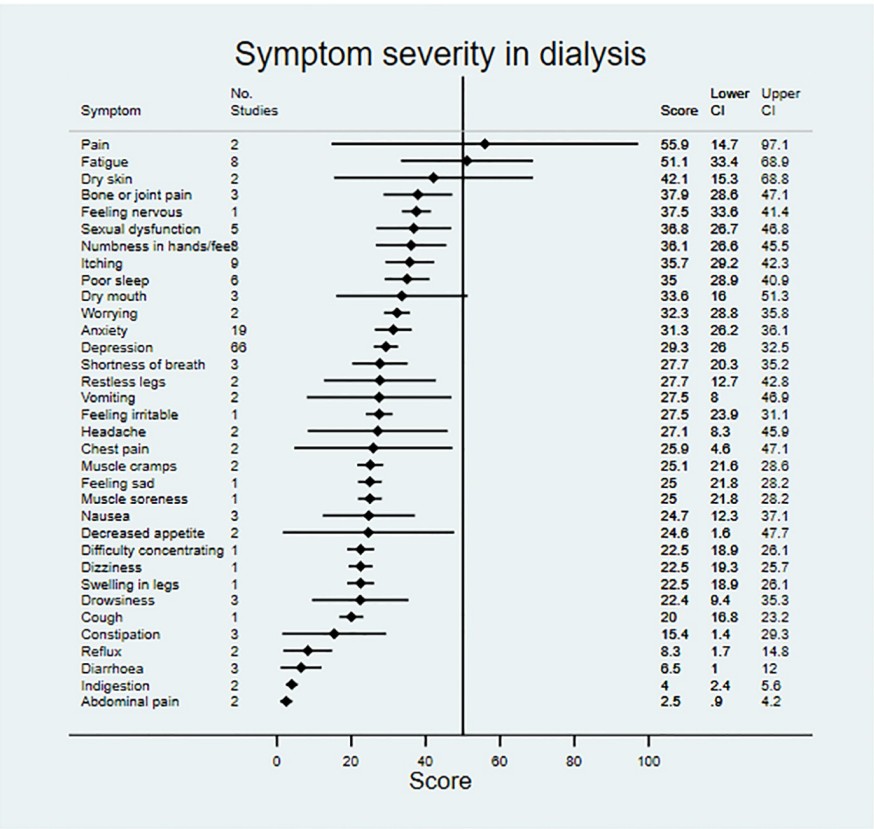

**Fig 6. Pooled symptom severity across included studies for patients on dialysis.** Scores represent mean severity scores converted to a 0–100 scale, where a higher score indicates greater severity; vertical line at 50 for reference.

numbness in hands/feet, and anxiety. No evidence of publication bias was found in meta-analyses of HRQOL data. All publication bias analyses are available in S6–S8 Appendices.

## Discussion

In this study, our first aim was to provide a global synthesis of symptom/HRQOL burden across all CKD stages. Overall, patients with CKD had a significantly increased symptom burden, and lower quality of life, compared to individuals without CKD. Patients reported a range of common and/or severe symptoms, with the precise configuration depending on the stage of CKD and RRT treatment modality received. Fatigue, however, was a very common and severe symptom in all patient groups. Symptom burden and quality of life were worst in patients receiving dialysis. In general, patients who had received a kidney transplant experienced fewer and less severe symptoms, and had an improved quality of life, compared to patients with CKD not receiving RRT or patients receiving treatment with dialysis. Transplantation, however, did not restore quality of life to levels seen in those without CKD.

Identification of the burden of symptoms for patients with advanced CKD is important: Many of the symptoms reported can be mitigated by changes in clinical management [19]. This synthesis will support clinicians and patients in consultations, to ensure that all potential symptoms associated with CKD are recognised, facilitating shared decision-making regarding management [20]. In addition, the data highlight key variations in symptom burden between

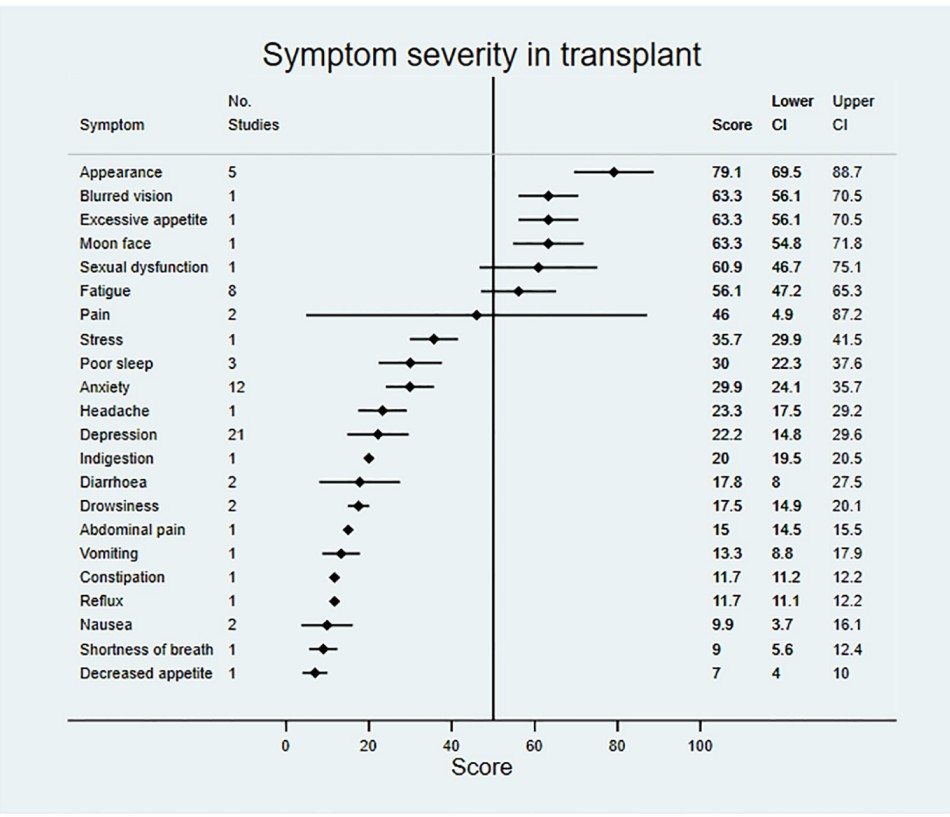

**Fig 7. Pooled symptom severity across included studies for patients in receipt of a kidney transplant.** Scores represent mean severity scores converted to a 0–100 scale, where a higher score indicates greater severity; vertical line at 50 for reference.

kidney disease modalities, allowing clinicians to identify fundamental differences and administer appropriate treatment. Moreover, the data may highlight key domains appropriate for inclusion within routine remote monitoring, a tool increasingly employed in clinical practice to support timely intervention in response to patient deterioration [8–11]. Recent reviews report on some symptoms, but do not quantify the symptom burden or accurately identify differences in symptom burden between treatment states [21,22]. The results of this study address this shortfall and provide information that has direct implications for clinical practice.

Our results concur with previous research regarding the prevalence of a number of symptoms, particularly in patients with stage 4/5 CKD. Almutary and colleagues conducted a 2013 review of symptom burden in CKD (stage 4/5 not on RRT/on dialysis), finding that the most prevalent symptoms were fatigue, feeling drowsy, pain, pruritus, and dry skin [21]. Murtagh et al. conducted a systematic review of the prevalence of symptoms in patients with end-stage renal disease, with the following prevalent symptoms highlighted: fatigue/tiredness, pruritus, constipation, anorexia, pain, sleep disturbance, anxiety, dyspnea, nausea, restless legs, and depression [22]. Differences between these studies and ours may be explained by our far larger sample size, resulting in much more precise estimates. Our study also has the advantage of comparing data across CKD clinical groups and including additional information regarding HRQOL and symptom severity.

Our second aim was to explore which symptom/HRQOL domains are modified by CKD and may be attributable to the disease. To the best of our knowledge, ours is the first study to

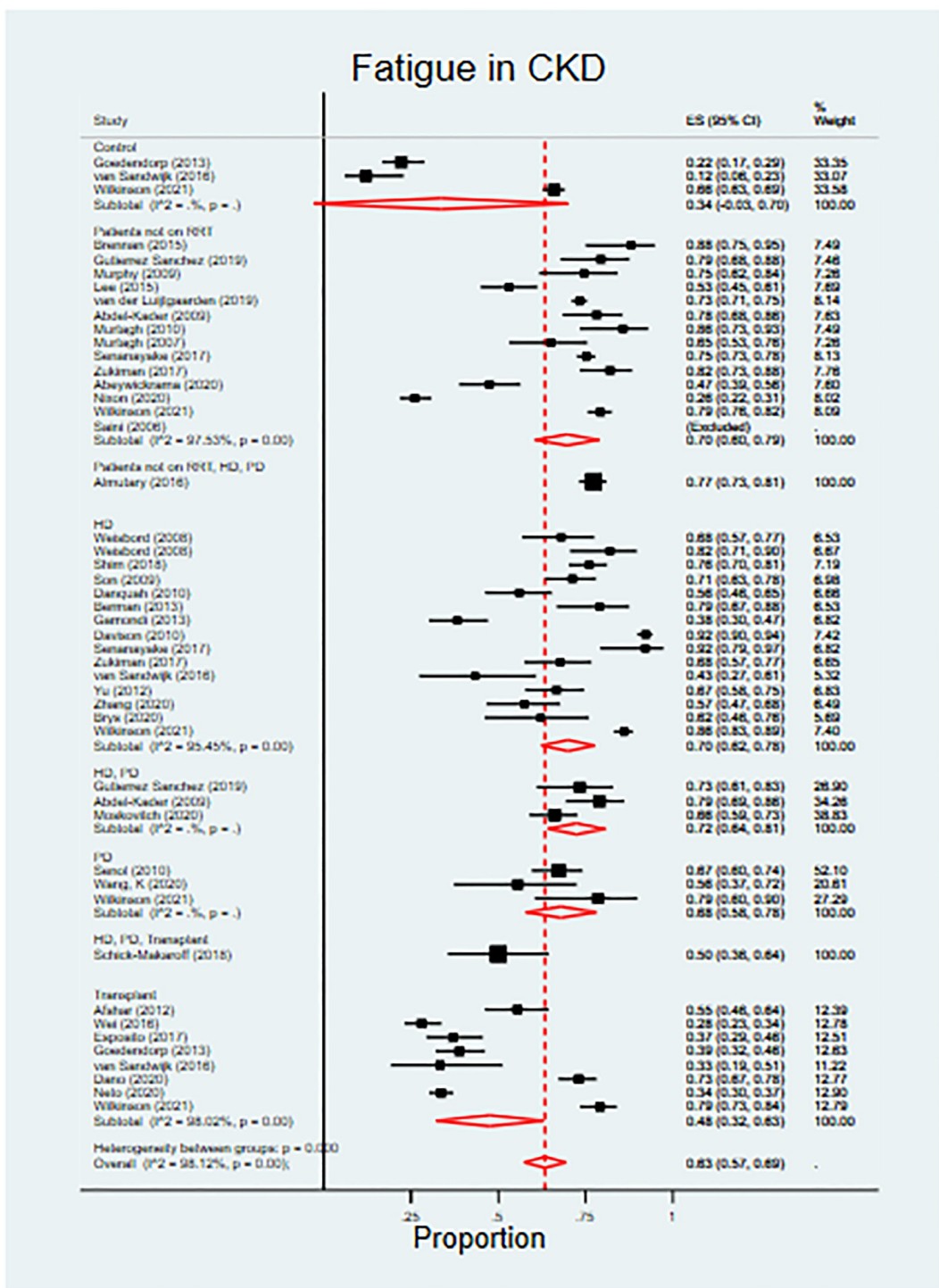

**Fig 8. Prevalence of fatigue in chronic kidney disease.** See S4 Appendix for full references for included studies. ES, effect size; HD, haemodialysis; PD, peritoneal dialysis; RRT, renal replacement therapy.

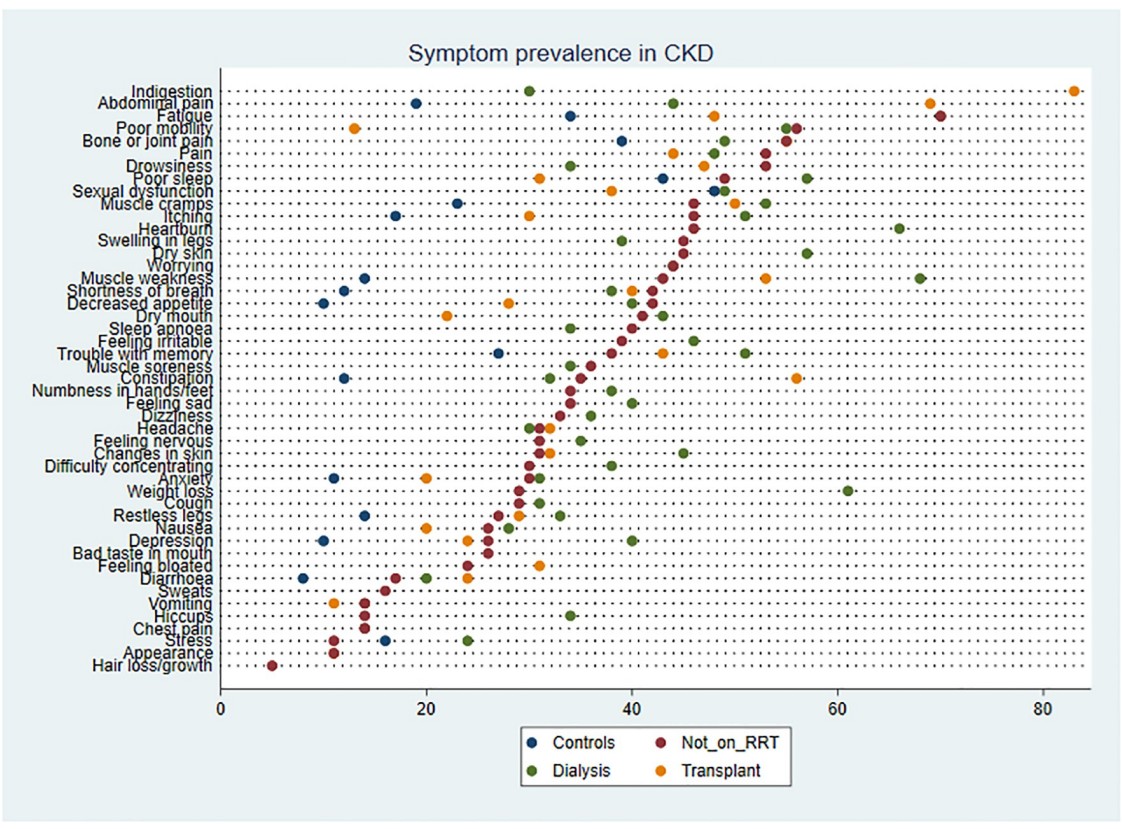

**Fig 9. Symptom prevalence comparison across groups.** CKD, chronic kidney disease; RRT, renal replacement therapy.

attempt this. Our results suggest there was a significantly lower prevalence of many symptoms in kidney transplant patients, compared to patients not on RRT and dialysis patients, which may suggest attribution; these symptoms included muscle weakness, fatigue, poor sleep, itching, decreased appetite, depression, dry mouth, and poor mobility. Collecting routine data on which symptoms/HRQOL are prevalent, impactful, and directly attributable to CKD is vital to improve understanding of individuals' experience of illness and to target treatment/support. This information can also be used alongside existing clinical data in discussions with patients to help better prepare them for CKD progression and to inform shared decisions around treatment [10]. We also found that some symptoms did not differ significantly across clinical groups and therefore may be largely unrelated to kidney function specifically, or the confidence intervals may preclude accurate interpretation. Further research may be required to explore the symptoms with broad confidence intervals, and hence greater uncertainty.

The third aim of the study was to determine which current PROMs may capture patient-important symptom/HRQOL information in a way that minimises patient burden. In total, 54 PROMs were used to collect data on symptoms across the included studies, and we found little consistency in the measures. This may be a consequence of the fact that no single tool measured >45% of symptoms reported in the population. This is problematic. At present, comprehensive measurement of symptoms would require that patients complete multiple PROMs, which may include large numbers of items or may have items that overlap. Such PROM 'item burden' has been widely recognised as an important threat to adherence [23]. Given the

**Table 3. Health-related quality of life outcomes in chronic kidney disease.**

| Measure | Not on RRT | Dialysis | Transplant |
|---|---|---|---|
| **Medical Outcomes Study 36-Item Short Form Health Survey (SF-36)** | | | |
| **Mental Component Summary*** | 66.6 (66.5–66.6) | 45.7 (45.5–45.8) | 50.0 (49.9–50.1) |
| Number of studies | 39 | 91 | 39 |
| Number of participants | 24,600 | 32,105 | 9,664 |
| **Physical Component Summary*** | 66.3 (66.2–66.4) | 35.5 (35.3–35.6) | 48.0 (47.9–48.1) |
| Number of studies | 39 | 91 | 39 |
| Number of participants | 24,600 | 32,105 | 9,664 |
| **Medical Outcomes Study 12-Item Short Form Health Survey (SF-12)** | | | |
| **Mental Component Summary*** | 49.8 (49.7–49.9) | 45.4 (45.2–45.6) | 48.2 (47.5–48.9) |
| Number of studies | 13 | 36 | 7 |
| Number of participants | 19,447 | 8,910 | 878 |
| **Physical Component Summary*** | 47.5 (47.3–47.6) | 35.2 (35.0–35.4) | 44.6 (44.0–45.3) |
| Number of studies | 13 | 36 | 7 |
| Number of participants | 19,447 | 8,910 | 878 |
| **Kidney Disease Quality of Life (KDQOL) instrument** | | | |
| **KDQOL Summary Score*** | 73.0 (73.0–73.2) | 64.6 (64.3–64.8) | 83.2 (84.8–87.5) |
| Number of studies | 19 | 69 | 10 |
| Number of participants | 30,689 | 12,222 | 1,374 |
| **Symptoms and problems*** | 85.9 (85.9–86.0) | 73.6 (73.4–73.8) | 86.1 (85.4–86.7) |
| Number of studies | 19 | 69 | 10 |
| Number of participants | 29,689 | 9,807 | 1,374 |
| **Effects of kidney disease*** | 71.7 (71.5–71.8) | 63.2 (63.0–63.5) | 87.5 (86.7–88.2) |
| Number of studies | 21 | 71 | 11 |
| Number of participants | 32,977 | 12,514 | 1,401 |
| **Burden of kidney disease*** | 50.6 (50.5–50.8) | 41.7 (41.4–42.0) | 72.0 (70.7–73.3) |
| Number of studies | 21 | 71 | 11 |
| Number of participants | 32,977 | 13399 | 1,401 |
| **EuroQoL–5 Dimension (EQ-5D)** | | | |
| **Index score^** | 0.882 (0.882–0.882) | 0.774 (0.767–0.781) | 0.840 (0.821–0.859) |
| Number of studies | 11 | 15 | 1 |
| Number of participants | 24,161 | 2,637 | 494 |

RRT, renal replacement therapy.

*Mean (95% CI) HRQOL score, 100 = best possible quality of life.

^Mean (95% CI) EQ-5D index score, 1 = best possible quality of life.

high number of symptoms experienced by patients with CKD, the use of contemporary psychometrics, encompassing item response theory (IRT) and computerised adaptive testing (CAT), may be warranted in order to develop new measures that capture sufficient information regarding all patient-important symptoms, whilst minimising questionnaire burden [24]. CATs efficiently select questions from an IRT-calibrated item bank that are targeted to an individual's ability/trait level using an adaptive algorithm, minimising the number of items administered, for example, the Patient-Reported Outcomes Measurement Information System (PROMIS) physical function CAT [25–27]. The findings of this systematic review will contribute to the construction of an item bank as part of the RCAT (Renal Computerized Adaptive Test) study [28].

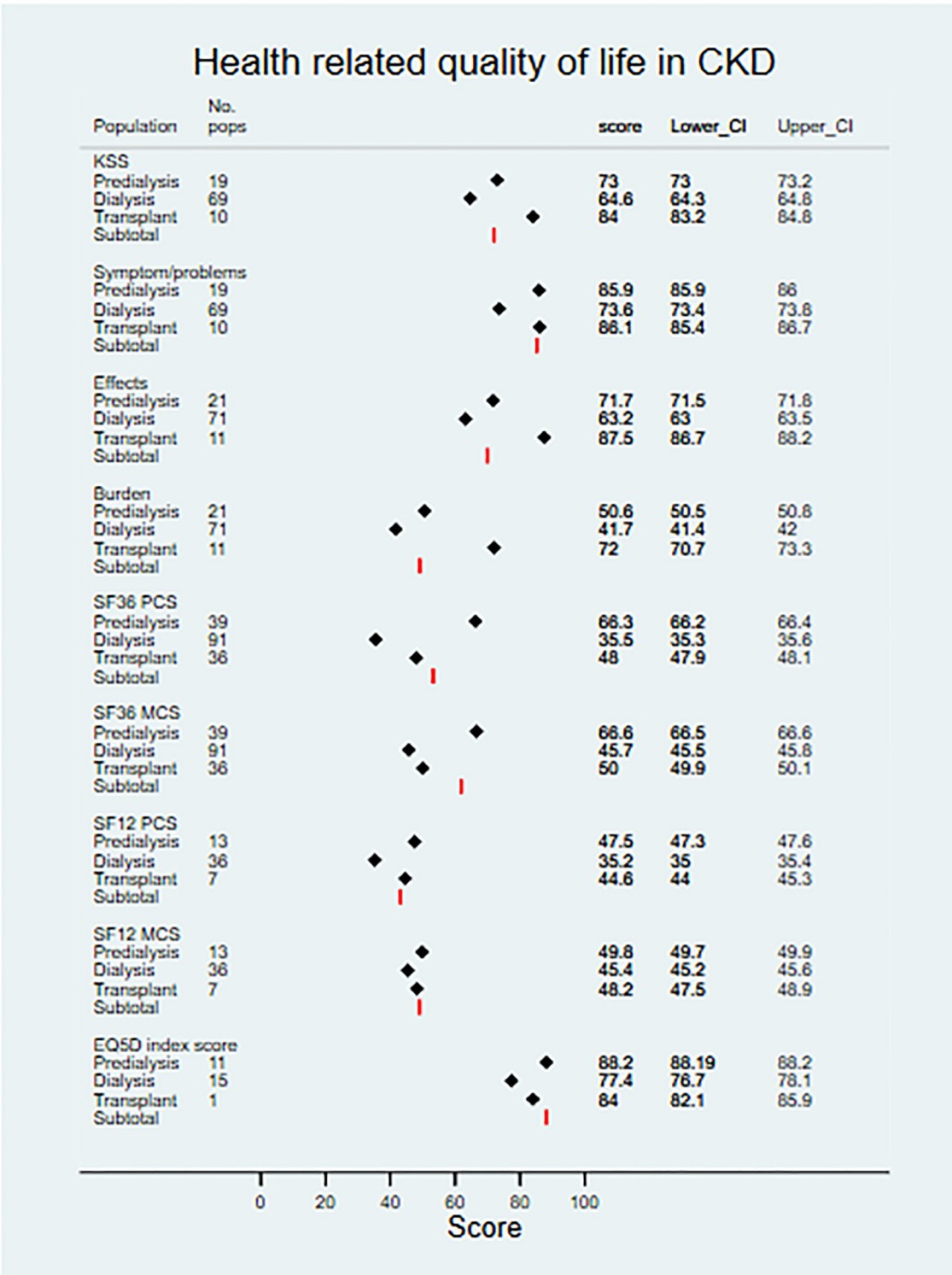

**Fig 10. Health-related quality of life in CKD.** CKD, chronic kidney disease; EQ5D, EuroQoL–5 Dimension; KSS, Kidney Disease Quality of Life Summary Score; MCS, Mental Component Summary; No. pops, number of populations; PCS, Physical Component Summary; SF12, 12-Item Short Form Health Survey; SF36, 36-Item Short Form Health Survey.

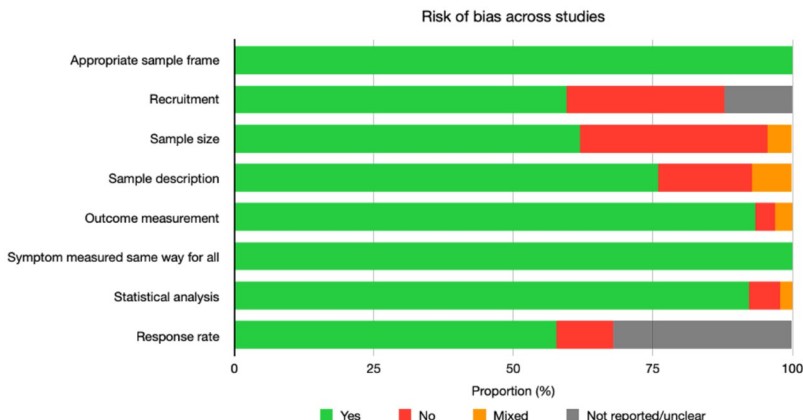

**Fig 11. Quality of included studies.** Yes = appraised as adequate; no = appraised as not adequate.

## Strengths and limitations

Our findings present a comprehensive overview of the differences in symptom and HRQOL prevalence and severity between patients with CKD stage 1–5 not on RRT, patients receiving dialysis, and transplant patients. In addition, where contemporaneous data were available, we were able to compare with controls. The study included >190,000 patients with CKD (from 62 countries) throughout the trajectory of the disease and its treatment. A large quantity of data was available from different settings, and this synthesis presents strong evidence for the ongoing and considerable impact of CKD on patients' lives. The review included data collected using PROMs only, hence data provided from CKD patients themselves. It is now widely understood that PROMs are patient centric, and provide information that is as important, if not more so, to patients than solely focusing on clinical outcomes [29–32].

Data were most frequently from cross-sectional studies, and whilst this is useful in providing an estimate of the prevalence and impact of symptoms at a population level, it does not address the day-to-day variation experienced by CKD patients and makes it challenging to draw robust conclusions around longitudinal patterns of symptom burden during the course of the disease.

A limitation is that we excluded non-English-language papers, meaning some potentially relevant studies may not have been included in our analysis. In addition, many symptom severity estimates came from single studies, and inconsistent use of PROMs across this literature hindered interpretation, especially with regard to clinical significance. This necessitated standardisation of severity scores onto a 0–100 scale to support meaningful synthesis [33]. Moreover, whilst there was information available in some of the studies on the severity of CKD, symptom burden with respect of excretory kidney function was limited, so the evolution of symptoms during the progression of CKD could not be assessed.

A further limitation was the considerable heterogeneity of included studies. This was not unexpected, as heterogeneity can be a common problem in systematic reviews of global prevalence data [12]. However, we followed established guidelines in our analysis [12], which suggest that in the presence of significant heterogeneity, random effects meta-analysis may be an appropriate method of generating a distribution that allows estimation of population differences with a quantifiable degree of probability.

### Future research

The findings of this review highlight several areas that warrant further research. In particular, additional high-quality studies exploring symptom severity are required in order to generate more precise estimates. We also found that there were fewer studies exploring symptom burden/HRQOL in the transplant population compared to other CKD groups. Finally, historical and ongoing use of many different symptom/HRQOL PROMs across studies poses particular challenges for those wishing to synthesise data. Future studies should focus on more consistent use of recommended outcome measures, such as those included in internationally endorsed core outcome sets [34,35], to facilitate comparisons between studies and enhance the generalisability of findings.

### Conclusion

This systematic review provides a detailed overview of the symptom/HRQOL profile across CKD clinical groups, with fatigue particularly prevalent, both in patients not on RRT and in those receiving dialysis. A number of symptoms were less prevalent and/or severe within the post-transplantation population, which may suggest attribution to CKD. HRQOL in patients with CKD was significantly worse than in individuals without the disease, particularly so in patients receiving dialysis. In general, patients receiving a transplant experienced lower symptom prevalence and severity and improved disease-specific quality of life, but this still did not reach the level of HRQOL of people without CKD. The findings of this review may support healthcare professionals when discussing, measuring, and managing the potential treatment burden associated with CKD.

## Supporting information

**S1 Appendix. PRISMA checklist.**
(DOC)

**S2 Appendix. Ovid MEDLINE search strategy.**
(DOCX)

**S3 Appendix. Information on included studies.**
(XLSX)

**S4 Appendix. List of included studies.**
(DOCX)

**S5 Appendix. List of excluded studies.**
(DOCX)

**S6 Appendix. Symptom prevalence across studies.**
(XLSX)

**S7 Appendix. Symptom severity across studies.**
(XLSX)

**S8 Appendix. Pairwise group comparisons.**
(XLSX)

**S9 Appendix. Review protocol.**
(DOCX)

**S10 Appendix. Summary of symptom outcome measures.**
(XLSX)

## Author Contributions

**Conceptualization:** Benjamin R. Fletcher, Sarah Damery, Olalekan Lee Aiyegbusi, Nicola Anderson, Melanie Calvert, Paul Cockwell, James Ferguson, Mike Horton, Muirne C. S. Paap, Chris Sidey-Gibbons, Anita Slade, Neil Turner, Derek Kyte.

**Data curation:** Benjamin R. Fletcher, Sarah Damery, Nicola Anderson, Derek Kyte.

**Formal analysis:** Benjamin R. Fletcher, Sarah Damery, Mike Horton, Anita Slade, Derek Kyte.

**Funding acquisition:** Olalekan Lee Aiyegbusi, Melanie Calvert, Paul Cockwell, James Ferguson, Mike Horton, Muirne C. S. Paap, Chris Sidey-Gibbons, Neil Turner, Derek Kyte.

**Investigation:** Benjamin R. Fletcher, Sarah Damery, Melanie Calvert, Paul Cockwell, Muirne C. S. Paap, Derek Kyte.

**Methodology:** Benjamin R. Fletcher, Sarah Damery, Nicola Anderson, Melanie Calvert, Paul Cockwell, Mike Horton, Muirne C. S. Paap, Anita Slade, Derek Kyte.

**Project administration:** Benjamin R. Fletcher, Derek Kyte.

**Resources:** Benjamin R. Fletcher.

**Supervision:** Melanie Calvert, Derek Kyte.

**Visualization:** Benjamin R. Fletcher, Derek Kyte.

**Writing – original draft:** Benjamin R. Fletcher, Sarah Damery, Olalekan Lee Aiyegbusi, Nicola Anderson, Melanie Calvert, Paul Cockwell, James Ferguson, Mike Horton, Muirne C. S. Paap, Chris Sidey-Gibbons, Anita Slade, Neil Turner, Derek Kyte.

**Writing – review & editing:** Benjamin R. Fletcher, Sarah Damery, Olalekan Lee Aiyegbusi, Nicola Anderson, Melanie Calvert, Paul Cockwell, James Ferguson, Mike Horton, Muirne C. S. Paap, Chris Sidey-Gibbons, Anita Slade, Neil Turner, Derek Kyte.

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
