## [Editor Report · Decision Letter 0]

25 Mar 2021

Dear Dr Fletcher, 

Thank you for submitting your manuscript entitled "Symptom burden and health-related quality of life in chronic kidney disease: a systematic review and meta-analysis." for consideration by PLOS Medicine.

Your manuscript has now been evaluated by the PLOS Medicine editorial staff and I am writing to let you know that we would like to send your submission out for external peer review.

Kind regards,

Caitlin Moyer, Ph.D.

Associate Editor

PLOS Medicine

---

## [Decision Letter · Decision Letter 1]

9 Aug 2021

Dear Dr. Fletcher,

Thank you very much for submitting your manuscript "Symptom burden and health-related quality of life in chronic kidney disease: a systematic review and meta-analysis." (PMEDICINE-D-21-01326R1) for consideration at PLOS Medicine. 

Your paper was evaluated by a senior editor and discussed among all the editors here. It was also discussed with an academic editor with relevant expertise, and sent to three independent reviewers, including a statistical reviewer. The reviews are appended at the bottom of this email and any accompanying reviewer attachments can be seen via the link below:

[LINK]

As you can see below, the reviewers have pointed out a number of concerns with the paper, including with the reporting of the methods and results. I am afraid that we will not be able to accept the manuscript for publication in the journal in its current form; however, we would consider a revised version that completely addresses the reviewers' and editors' comments. Obviously we cannot make any decision about publication until we have seen the revised manuscript and your response, and we plan to seek re-review by one or more of the reviewers. 

We expect to receive your revised manuscript by Aug 30 2021 11:59PM. Please email us (plosmedicine@plos.org) if you have any questions or concerns.

We look forward to receiving your revised manuscript. 

Sincerely,

Caitlin Moyer, Ph.D.

Associate Editor 

PLOS Medicine

plosmedicine.org

1. Please completely address all reviewer comments, particularly the concerns mentioned by Reviewer 2 regarding reporting of the data extraction and analysis methods, and assessment of heterogeneity, including how the high degree of heterogeneity impacts pooling across studies in terms of prevalence and severity, and the interpretation of the meta analyses.

2. Please update the search to the present time (currently through February of 2020 and updated through October 2020).

3. Abstract: Please report your abstract according to PRISMA for abstracts, following the PLOS Medicine abstract structure (Background, Methods and Findings, Conclusions) http://www.plosmedicine.org/article/info:doi/10.1371/journal.pmed.1001419 .

Please combine the Methods and Findings sections into one section, “Methods and findings”. Please provide the dates of search, types of study designs included, exclusion criteria including language exclusions, and synthesis/appraisal methods. Please mention how you evaluated study quality and risk of bias, including publication bias.

4. Abstract: Methods and Findings: Please present quantitative data in support of the main findings of the meta analysis. As the last sentence of the Methods and Findings, please describe the main limitation(s) of the study's methodology.

5. Author summary: At this stage, we ask that you include a short, non-technical Author Summary of your research to make findings accessible to a wide audience that includes both scientists and non-scientists. The Author Summary should immediately follow the Abstract in your revised manuscript. This text is subject to editorial change and should be distinct from the scientific abstract. Please see our author guidelines for more information: https://journals.plos.org/plosmedicine/s/revising-your-manuscript#loc-author-summary

6. Main text: For citations of references in the text, please place reference numbers within square brackets placed before the sentence punctuation. For multiple references, please do not include spaces within brackets.

7. Main text: Please include line numbers with the revised version.

8. Methods: Thank you for noting that your study protocol was registered. Please note whether the protocol and analysis methods were developed prospectively. Please make sure that the Methods section transparently describes when analyses were planned, and when/why any data-driven changes to analyses took place.

9. Methods: Please provide a list of excluded studies and reasons for exclusion.

10. Methods: Please note how publication bias was assessed.

11. Results: As mentioned by Reviewer 1, please include a table with each study summarizing the design, instruments used, outcome measures, size, etc. and include quality assessment and risk of bias for each.

12. Results: Please provide an assessment of how results differ by excluding studies judged to be at high risk of bias.

13. Results: Meta-analysis for fatigue prevalence across CKD levels: Please provide both p values and 95% CIs for comparisons of fatigue prevalence between groups, in the text as well as in Figure 3.

14. Discussion: Please present and organize the Discussion as follows: a short, clear summary of the article's findings; what the study adds to existing research and where and why the results may differ from previous research; strengths and limitations of the study; implications and next steps for research, clinical practice, and/or public policy; one-paragraph conclusion.

15. Figure 4: Please provide an X axis label. The individual symptoms are difficult to read on the left, please increase the font size.Please provide the meta analysis results for each symptom shown in Figure 4 as displayed for Figure 3.

16. PRISMA diagram: Please note the “other sources” for the 8 additional records identified. Please note the reasons for records excluded at each stage.

17. PRISMA Checklist: When completing the checklist, please use section and paragraph numbers, rather than page numbers.

Comments from the reviewers:

Reviewer #1: The stated aim of this review is to produce a comprehensive and consolidated synthesis of symptom prevalence/severity and HRQOL across CKD stage and treatment groups.

Comments:

"The Preferred Reporting Items for Systematic Reviews and Meta-Analyses (PRISMA) guidelines were followed (PRISMA checklist, supplementary appendix 1) and the study protocol was registered with PROSPERO (CRD42020164737)."

The authors have suitably provided the PRISMA checklist in the supplementary material. 

Can they also please provide a copy of the study protocol in the supplementary material?

"Three authors (BF, SD, DK) independently assessed selected articles for eligibility at the title/abstract and full-text stages."

Can the authors please confirm if the articles were triple checked, or were they divided up between the three assessors?

Were there any cases of disagreement? If so, how were these resolved?

Also, were the authors of similar expertise and experience?

"The following data were independently extracted into a pre-piloted spreadsheet by three authors (BF, SD, DK)..."

As for the previous comment, can the authors please provide similar clarifications for the data extraction process?

"Symptom prevalence data, and mean HRQOL scores were combined using random-effects meta-analysis. To aid comparison of symptom severity data provided across different outcome measures, all mean severity scores were converted to a 0-100 scale, where a higher score indicates greater severity. For HRQOL scores, 100 represents best possible quality of life. Symptom severity scores were also combined using random effects meta-analysis. A weighted composite summary score (Kidney Summary Score or KSS) for the Kidney Disease Quality of Life (KDQOL) instrument was calculated by combining the 'Symptom/Problems', 'Effects' and 'Burden' domains, using the method reported by Peipert et al. (2019). Exploratory subgroup analysis was used to compare prevalence and score (severity and HRQOL) estimates between groups in meta-analyses (predialysis v dialysis, predialysis v transplant and dialysis v transplant). To account for multiple testing, sharpened False Discovery Rate (FDR) q-values were computed, and adjusted p-values are reported."

and

"Random effects meta-analyses were used to pool data on symptom prevalence and severity, and HRQOL scores. We stratified meta-analyses by CKD status: Stage 1-5 not on dialysis (predialysis), on dialysis, and post renal transplantation."

The authors have applied a technically appropriate and rigorous statistical methodology.

Can the authors please discuss heterogeneity within the methods, results and conclusions?

It is noted that I2 in Figure 3 is > 95%, which is substantially high and may mean the results of the meta-analysis are misleading.

Can the authors confirm whether Figure 6 and Table 4 show the same thing?

If so, it is suggested to remove one of these to avoid duplication.

It is also suggested to replace the List of Included studies in Appendix 3, with a table describing each (type, size, dates, location etc.) study with its associated JBI quality assessment.

Reviewer #2: Thank you for the opportunity to review this manuscript describing the symptom burden and health-related quality of life in CKD using a systematic review and meta-analysis of the published literature 2000 to Feb 2020. The authors used three main search engines to retrieve the publications (MEDLINE, PschINFO, CINAHL), which yielded 415 studies to include in the review and meta-analysis. This is an interesting number since only 4 years before, a systematic review published in PLOS One (https://journals.plos.org/plosone/article?id=10.1371/journal.pone.0179733) only identified 66 publications using an additional search engine (EMBASE) that covered the same patient-reported outcome measures in CKD. I have major concerns regarding the methodology that was used - particularly for meta-analyses, largely regarding pooling of the data across study populations that used different study instruments without formal assessment of heterogeneity that would indicate if it would be worthwhile to pool the results. The authors already note in that the outcome measures (pg 8) there was "little consistency across measures". The methods section is very under developed and the reader cannot clearly understand the methods used. Additionally, I have concerns about description of prevalence and severity of symptoms as the true denominator is unknown and is affected by publication bias and ascertainment bias (e.g., more studies and instruments asked about pain and fatigue rather than itching - does that mean that itching is less important to the patients?). The discussion is also under-developed. I think the paper would be strengthened by reporting the number of studies/instruments that included particular symptom measures, what the typical reported level of that symptom is across stages of CKD (if possible) without formal analyses since the ability to pool is likely not sufficient. 

Throughout manuscript - please use alternative term for "pre-dialysis" - for example, non-dependent CKD as not all patients with CKD will go on to develop dialysis

Abstract

- Background: clarify what "treatment groups" means and what does inform capture mean?

- Methods: indicate end year of search

- Conclusion: why only highlight the systematic review portion

Introduction

- Spelling error for vulnerable

- 

Methods:

- Why was EMBASE not included?

- Formal tests of heterogeneity?

- Data analysis section - underdeveloped and need more elaboration of how the studies were pooled and how the instruments were harmonized for pooling - for example - if there was any mention of fatigue in the PROM - was it "counted" as a measure of fatigue? - again how was severity of that measure scored?

- need to also describe how mean HRQOL scores were determined across different instruments - were these also standardized?

- How was a severity score determined?

- How was the weighted composite summary score constructed - need more elaboration than a prior citation

- Suggest that instead of exploratory meta-analyses comparing scores across stages of CKD - first describe the differences

Figure 1 - I think they are misleading - I think is affected by publication bias - is it really symptom prevalence or prevalence of symptoms asked prior studies? 

Figure 2 - it very difficult to interpret these without learning more about the severity scores

Figure 3 - what does this add to Figures 1?

Figure 4 - as above for figure 2 - need more description about severity scores

Table 3- was individual level used to pool these results or are they from averages (means) reported in the published papers?

Trends in prevalence (page 21)

- The authors note that "data from non-CKD control populations are sparse" which is not possible as there are many many studies that have used legacy PROMS, including SF-36 in non-CKD populations that can provide normative data for comparison https://hqlo.biomedcentral.com/articles/10.1186/s12955-017-0625-9

Discussion

- I think it is misleading to report there was lower prevalence of symptoms in transplant compared to non-dialysis CKD or dialysis since the denominator differs so much, there were fewer studies and the outcome measures could have asked fewer questions

- On page 29 - "we found little consistently in the PROMS used to capture data around symptoms in CKD" - how do the authors justify meta-analyzing the data?

- Suggest the authors include the following citation in their writing about the use of IRT and CAT to develop new measures - as this has been developed by the PROMIS program using mixed methods - https://www.healthmeasures.net/explore-measurement-systems/promis

- The introduction provides background on how this will inform ePROs - I didn't see further discussion on this - which surveys collected were collected with digital/electronic means?

Reviewer #3: Thank you for the opportunity to review this paper.

It has clearly involved a considerable amount of work to synthesise the mass of data, for which the authors deserve credit.

My main comments concern the framing of the work, some commentary of the methods and then a view on the conclusions reached. This is an important area that requires focus.

The introduction produces the background and rationale for the work but the section on electronic reporting seemed redundant. It was not part of the study. It conflates the definitions of symptoms and the tools used with the operational method to collect those tools. So it did not seem to add anything to the objectives or the methodology.

The introduction lays out three clear objectives for the work 'The aim of this study was to: (i) produce a comprehensive and consolidated synthesis of symptom prevalence/severity and HRQOL across CKD treatment groups; (ii) explore which symptoms/HRQOL domains are modified by CKD and may be attributable to the disease; (iii) determine which PRO measures (PROMs) are currently available to capture symptom prevalence/severity and HRQOL in CKD.'

It was not clear how these objectives were achieved from the discussion. It would be useful to reflect back those challenges as a summary of the study.

Turning to the data itself, there was some discussion on control groups but again it was difficult to pick out the narrative of symptom evolution from normality through to CKD (and across the stages of CKD) to renal replacement therapy (dialysis and transplantation). Figure 4 helps in one sense (and is the fatigue line missing the mark for the dialysis population?) but it only covers prevalence. The burden comes from both prevalence and from severity. A discussion on this two dimensional view of symptom burden would be valuable. That would provide insight into the utility of measures as drivers to improve care for people with kidney disease.

A minor note on the methods - it was unclear whether those reviewing the papers did so independently and then a consensus was reached where needed. There is also a typo on manuscript page 3 line 10.

[LINK]

---

## [Decision Letter · Decision Letter 2]

11 Jan 2022

Dear Dr. Fletcher,

Thank you very much for submitting your manuscript "Symptom burden and health-related quality of life in chronic kidney disease: a global systematic review and meta-analysis." (PMEDICINE-D-21-01326R2) for consideration at PLOS Medicine. 

Your revised paper was evaluated by a senior editor and discussed among all the editors here. It was also discussed with an academic editor with relevant expertise, and sent to two of the original reviewers, including a statistical reviewer. The reviews are appended at the bottom of this email and any accompanying reviewer attachments can be seen via the link below:

[LINK]

Giving the remaining points noted by one reviewer, I am afraid that we will not be able to accept the manuscript for publication in the journal in its current form, but we will consider another revised version that addresses the reviewers' and editors' comments. Obviously we cannot make any decision about publication until we have seen the revised manuscript and your response, and we plan to seek re-review by one or more of the reviewers. 

We expect to receive your revised manuscript by Feb 01 2022 11:59PM. Please email us (plosmedicine@plos.org) if you have any questions or concerns.

We look forward to receiving your revised manuscript. 

Sincerely,

Caitlin Moyer, Ph.D.

Associate Editor

PLOS Medicine

plosmedicine.org

1. Please completely address the remaining points of Reviewer 2.

2. Abstract: Methods and Findings: Please explicitly describe the participants/comparator groups among the patients with CKD included.

3. Abstract: Methods and Findings: Please mention the types of/any restrictions on design of studies included (e.g. longitudinal, cross-sectional).

4. Abstract: Line 43-44: Please mention how (the specific tools used) methodological quality and risk of bias were assessed.

5. Abstract: For the main outcomes described, it would be helpful to note the numbers of studies/patients included for each.

6. Abstract: Line 49: Please define the abbreviation “RRT” at first use. 

7. Abstract: Line 51-52: Please mention the comparison group for this analysis, and please report 95% CIs and p values for each symptom mentioned.

8. Abstract: Lines 54-58: Please also report the p values for these comparisons.

9. Methods: Line 124-125: Please reference the supporting information file that contains the study protocol.

10. Methods: Please mention the types of studies/designs included. The search did not include non-English language sources of studies (45 studies excluded for this reason). Please acknowledge and discuss this as a limitation.

11. Methods: Please provide complete details for the fixed and random effects models used. Please indicate for which analyses fixed vs. random effects models were used.

12. Discussion: Line 381, Line 461: Please temper claims of primacy with “To the best of our knowledge” or similar.

13. Table 1: Please note that “Population” indicates stage of CKD. Please indicate in the table that values are number of studies and (%) if accurate. It may be helpful to include a legend for the table, defining all abbreviations (RRT) in the legend.

14. Table 3: Please clarify the values presented in the table.

15. Figure 4: Please include a legend describing all abbreviations used. 

16. S3 Appendix, S6 appendix, S7 appendix: Please include a legend for this table, defining all abbreviations and color-coding used. We suggest choosing a color-coding scheme that avoids the need to distinguish red and green.

17. S8 Appendix: Separate titles and legends for each item (each tab) in this file would be helpful to understand which comparisons are being shown and what is illustrated in each table.

18. PRISMA Checklist: Thank you for including the PRISMA checklist. Please use section and paragraph numbers to refer to locations within the text. Please do not use page numbers or line numbers.

Comments from the reviewers:

Reviewer #1: The authors have satisfactorily responded to each comment in turn, amending the manuscript as required.

Reviewer #2: Thank you for the opportunity to review this revised manuscript describing the symptom burden and health related quality of life in CKD. 

I do think this revision is improved but is not yet ready for publication. The authors have addressed many of my concerns but did not directly respond to the second paragraph of text of my original review. Some concerns were addressed in separate comments but I am still concerned that the discussion is largely underdeveloped. It lacks synthesis and interpretation of the study's findings and largely repeats the study's findings, which is particularly apparent in the second paragraph of the discussion. The third paragraph of the discussion also contains too many details about the cited study - please only highlight main similarities. The entire discussion could be strengthened by including more interpretation.

Other concerns

- Methods: I am concerned about the psychometric validity of rescoring the the self-reported scores to 100. Are there any previous studies or citations that have reported rescaling of the scores to 100? 

- Figures - please clarify what the word prevalence indicates ? Is it the % of those included in each of the studies that reported the symptom? Please provide that explanation in the Figure legend.

[LINK]

---

## [Decision Letter · Decision Letter 3]

14 Feb 2022

Dear Dr. Fletcher,

Thank you very much for re-submitting your manuscript "Symptom burden and health-related quality of life in chronic kidney disease: a global systematic review and meta-analysis." (PMEDICINE-D-21-01326R3) for review by PLOS Medicine.

I have discussed the paper with my colleagues and the academic editor and it was also seen again by one of the reviewers. I am pleased to say that provided the remaining editorial and production issues are dealt with we are planning to accept the paper for publication in the journal.

[LINK]

We look forward to receiving the revised manuscript by Feb 21 2022 11:59PM.   

Sincerely,

Caitlin Moyer, Ph.D.

Associate Editor 

PLOS Medicine

plosmedicine.org

Requests from Editors:

1. Title: Please capitalize the first word of the subtitle, please remove the punctuation after “meta-analysis” and please be sure to update the title in the manuscript submission system: “Symptom burden and health-related quality of life in chronic kidney disease: A global systematic review and meta-analysis”

2. Abstract: Methods and Findings: Please report the p values in addition to the 95% CIs for quality of life measures compared between patients on dialysis and patients not on RRT, or those who received kidney transplant, rather than reporting as p<0.01.

3. Throughout: In-text reference call-outs: Please include a space between the end of the preceding word and the bracketed reference number, for example [1].

4. Results: Lines 373-376: Please provide a reference in the text to where the data of exploratory analyses comparing HRQOL scores between populations can be found.

5. Discussion: Line 450: Please replace “compliance” with “adherence” in this sentence.

6. Discussion: Line 459: We suggest removing the URL from the text, and including the website in the reference list.

7. Discussion: Please slightly re-organize the discussion as follows. After the discussion of strengths and limitations of the study, please discuss the implications and next steps for research, clinical practice, and/or public policy, followed by your one-paragraph Conclusion.

8. Lines 508-542: Please remove the “Funding and Conflicts of Interest” section from the main text, and please be sure all information is complete and accurately entered into the manuscript submission system.

9. Figure 1: Please define all abbreviations (PROM, CKD) in the legend.

10. Figure 2: Please define all abbreviations (CKD, RRT) in the legend.

11. Figure 3 (all parts): A short description of the symptom severity score would be helpful in the legend. Please also explain the vertical line at the score of 50. Please define all abbreviations (RRT, CKD) in the legend.

12. Figure 4: It would be helpful if the categories could be made to stand out more easily (Control, Patients not on RRT, etc. ).Please define all abbreviations used in the legend.

13. Figure 5: Please define all abbreviations used (CKD, RRT) in the legend.

14. Figure 6: Please change “lowerci” and “upperci” to Lower CI/Upper CI to match the formatting in other tables. In the legend, please define all abbreviations used. Please define all abbreviations (CKD, RRT) in the legend.

15. Figure 7: In the legend, please clarify if “yes” and “no” correspond to the question of risk of bias, or with the adequacy appraisal (being reported as adequate or not).

16. Table 1: Please note that the full list of countries represented is not included. We suggest including the full list, if not in Table 1, please include such a list and please mention where a full list of the 62 countries may be found.

17. Table 2: Please provide some additional context for the different PROM described in the legend. Please define “RRT” in the legend.

18. Table 3: Please define all abbreviations (e.g. HRQOL, CKD) in the legend.

19. We suggest providing a short descriptive legend for each Supporting Information Table.

20. S3 Appendix: Please define all abbreviations used in a legend for the Table.

21. S8 Appendix: Please provide a descriptive legend for each tab in the file, including defining all abbreviations used and clarifying the values presented in red for prevalence and severity quartile tabs.

Comments from Reviewers:

Reviewer #2: This revision is much improved and adequately addressed my prior concerns. There are multiple spelling errors throughout the manuscript. For example, "clincial" on page 34, "acurate" on page 35. "Finaly" on page 36. Please carefully correct any spelling/grammatical errors. 

Additionally, please clarify what "latter group" on the beginning of page 35 refers to. Please ensure each Table and Figure has footnote definition for RRT and HRQOL, as appropriate.

[LINK]

---

## [Editor Report · Decision Letter 4]

23 Feb 2022

Dear Dr Fletcher, 

On behalf of my colleagues and the Academic Editor, Sanjay Basu, I am pleased to inform you that we have agreed to publish your manuscript "Symptom burden and health-related quality of life in chronic kidney disease: A global systematic review and meta-analysis." (PMEDICINE-D-21-01326R4) in PLOS Medicine.

Please also address the following editorial requests:

-S3 Appendix, S6 Appendix, S8 Appendix, and S10 Appendix: We suggest moving the descriptive legends to a fixed location of the file (a row just below the title, for example) rather than placing it within a text box over the table.

-S8 Appendix: In the legend, please define “predialysis” as those not on RRT (similar to the main text). Please include legends for “Prevalence quartiles” and “Severity quartiles” tabs, including an explanation for the values in red font.

PRESS

Sincerely, 

Caitlin Moyer, Ph.D. 

Associate Editor 

PLOS Medicine